# Monosynaptic *trans*-collicular pathways link mouse whisker circuits to integrate somatosensory and motor cortical signals

**Jesús Martín-Cortecero**\*[☯], **Emilio Ulises Isaías-Camacho**[☯], **Berin Boztepe**[¤☯], **Katharina Ziegler, Rebecca Audrey Mease, Alexander Groh**[ORCID]\*

Medical Biophysics, Institute for Physiology and Pathophysiology, Heidelberg University, Germany

[☯] These authors contributed equally to this work.
[¤] Current address: Neuroradiology Department, University Hospital Heidelberg, Heidelberg, Germany
\* martin-cortecero@physiologie.uni-heidelberg.de (JMC); groh@uni-heidelberg.de (AG)

**Data Availability Statement:** All relevant data are within the paper and its Supporting Information files. Data files for the figures and electrophysiology data are available from the public

## Abstract

The superior colliculus (SC), a conserved midbrain node with extensive long-range connectivity throughout the brain, is a key structure for innate behaviors. Descending cortical pathways are increasingly recognized as central control points for SC-mediated behaviors, but how cortico-collicular pathways coordinate SC activity at the cellular level is poorly understood. Moreover, despite the known role of the SC as a multisensory integrator, the involvement of the SC in the somatosensory system is largely unexplored in comparison to its involvement in the visual and auditory systems. Here, we mapped the connectivity of the whisker-sensitive region of the SC in mice with *trans*-synaptic and intersectional tracing tools and in vivo electrophysiology. The results reveal a novel *trans*-collicular connectivity motif in which neurons in motor- and somatosensory cortices impinge onto the brainstem-SC-brainstem sensory-motor arc and onto SC-midbrain output pathways via only one synapse in the SC. Intersectional approaches and optogenetically assisted connectivity quantifications in vivo reveal convergence of motor and somatosensory cortical input on individual SC neurons, providing a new framework for sensory-motor integration in the SC. More than a third of the cortical recipient neurons in the whisker SC are GABAergic neurons, which include a hitherto unknown population of GABAergic projection neurons targeting thalamic nuclei and the zona incerta. These results pinpoint a whisker region in the SC of mice as a node for the integration of somatosensory and motor cortical signals via parallel excitatory and inhibitory *trans*-collicular pathways, which link cortical and subcortical whisker circuits for somato-motor integration.

## Introduction

The superior colliculus (SC) is part of a phylogenetically ancient brain network that directs quick motor actions in response to ascending sensory signals [1,2]. As such, the SC is a central hub for multiple sensory-motor arcs linking sensory information to motor actions. From an

repository heiDATA: https://doi.org/10.11588/data/DNOSZG.

**Funding:** This work was supported by the German Research Foundation (DFG Grants GR3757/3-1, GR3757/4-1 to AG), the Heidelberg Graduate Academy completion grant through the Landesgraduiertenförderung program with funds allocated by the German Ministry of Science, Research and Arts (salary grant to EIC), the Chica and Heinz Schaller Stiftung (salary grant to RM) and the Brigitte-Schlieben-Lange Programm by the Ministry of Science, Research and the Arts Baden-Württemberg (salary grant to RM). We acknowledge the data storage service SDS@hd and high-performance computing initiative bwHPC, supported by the Ministry of Science, Research and the Arts Baden-Württemberg (SDS@hd and bwHPC) and the German Research Foundation (DFG) through grant INST 35/1314-1 FUGG and INST 35/1503-1 FUGG (SDS@hd). The funders had no role in study design, data collection and analysis, decision to publish, or preparation of the manuscript.

**Competing interests:** The authors have declared that no competing interests exist.

**Abbreviations:** BC, barrel cortex; Bs, brainstem; IQR, interquartile range; iRN, inhibitory recipient neuron; LP, lateral-posterior nucleus; LSC, lateral SC; MC, motor cortex; RAC, relative axon count; RN, recipient neuron; SC, superior colliculus; VM, ventro-medial; ZI, zona incerta.

evolutionary perspective, much of the SC's function in mammalian brains has been taken over by the neocortex, via parallel, cortically controlled sensory-motor arcs [3]. Intriguingly, recent work in the visual and auditory systems demonstrate that SC-mediated behaviors are modulated by cortical inputs [4–6], raising the question of how cortex- and SC-mediated sensory-motor arcs interact to organize desirable behavior. More specifically, it is not well understood how cortico-collicular pathways engage with SC microcircuitry and, in turn, generate SC output signals to the brainstem and diencephalon (including thalamus). This information is essential not only to understand how cortico-collicular pathways may coordinate between the "new" cortical and the "old" collicular arcs, but also to answer principal questions concerning the identity of cortical-recipient (i.e., targeted by cortex)–SC neurons: to where do they project, are they excitatory or inhibitory, and do they receive sensory and/or motor signals? SC circuits have mostly been studied in relation to vision [2,6–8], i.e., looking at functions of the "visual SC." In contrast, somatosensory functions of the SC are less studied and less understood, even though somatosensory functions such as whisker sensation are vital for rodents and other animals [9–12]. This study set out to address the involvement of "somatosensory SC" circuits in the whisker system and focused on 3 principal questions about the organization of *trans*-collicular pathways that mediate cortical input to SC downstream targets (Fig 1).

(1) To where do cortical-recipient SC neurons project? Cortico-collicular signals may be routed directly to SC downstream targets (i.e., in the diencephalon and brainstem) via monosynaptic *trans*-collicular pathways and/or more indirectly, via intracollicular connections to SC output neurons [2] (Fig 1, left panel). While intracollicular circuits are well described [13], evidence for monosynaptic *trans*-collicular pathways is limited because relatively few studies have delineated the precise input–output connectivity of defined subpopulations in SC [2,14]. Indeed, input–output connectivity of the SC is mostly considered at the level of SC layers [2,15] and radial cortico-collicular input zones [16]. However, these mesoscopic modules comprise intermingled populations of input and output neurons as well as input neurons with different input identities [16–18], for example, retinal and cortical inputs [19]. Therefore, dissection of input-defined cell types and analysis of their outputs is critical to unambiguously link specific SC input pathways to specific SC downstream targets and to determine whether cortical input signals can be transformed directly into SC output to downstream SC target circuits.

(2) Are cortical-recipient SC neurons excitatory or inhibitory? The SC comprises excitatory and inhibitory neurons; for example, in the visual SC, approximately one-third of the neurons are GABAergic [15,20]. However, very little is known about GABAergic neurons in the somatosensory SC (Fig 1, center panel), including their proportions, their inputs and outputs,

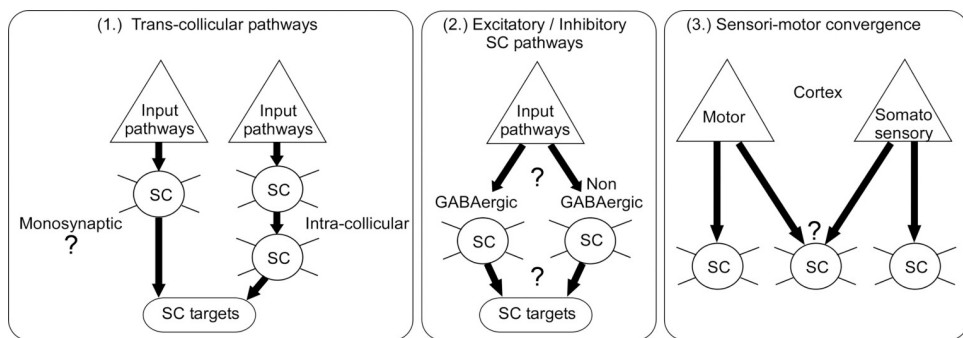

**Fig 1. Putative synaptic connectivity motifs for the transfer of cortical information to SC downstream targets via single SC neurons.**

and whether they are classic local interneurons or projections neurons. Determining how cortical pathways engage specific excitatory or inhibitory SC circuits and identifying which downstream pathways emanate from these circuits is indispensable to make circuit-mechanistic predictions about recently discovered top-down modulation of SC-mediated innate escape and defense behaviors [4–6].

(3) Do pathways from motor and somatosensory cortices converge at the level of individual SC neurons? The multisensory nature of the SC has motivated a large body of studies demonstrating the integration of visual, auditory, somatosensory, and motor signals in the SC [17,18,21–25]. Nevertheless, whether sensory and motor cortex pathways converge in individual SC neurons remains untested (Fig 1, right panel). The lateral SC (LSC) has been identified as a putative point of convergence of somatosensory and motor cortical efferents [16], making it a promising target to test this hypothesis.

Here, we mapped the input–output connectivity of the mouse SC on the single-cell level with *trans*-synaptic tracing, intersectional viral approaches, and optogenetic-assisted electrophysiology. In awake animals, we identify a whisker-sensitive region in the SC, which receives trigeminal and cortico-collicular input from motor and somatosensory cortex. We then targeted specific subsets of input-defined pathways from the brainstem and motor and somatosensory cortices to the SC and traced their axonal outputs to downstream targets, revealing direct *trans*-collicular pathways, which provide disynaptic long-range links between cortical pathways and SC target regions. *Trans*-synaptic labeling in combination with optogenetic input mapping reveals long-range input convergence on the level of individual SC neurons, with approximately one-third of the cortical recipient SC neurons receiving convergent input from both motor and somatosensory cortex. We find that long-range input pathways extensively target inhibitory SC neurons, which, in turn, give rise to long-range GABAergic projections to thalamic nuclei and the zona incerta (ZI). In sum, this study pinpoints a "whisker SC," in which converging cortical and brainstem inputs innervate GABAergic and non-GABAergic SC neurons, which, in turn, provide parallel *trans*-collicular pathways to downstream whisker circuits in the diencephalon and brainstem. These results suggest that this cortical control of SC directly affects the brainstem-SC-brainstem sensory-motor arc, as well as the outputs from SC to diencephalic stations.

## Results

### The whisker-sensitive LSC receives input from the motor cortex and barrel cortex and from the brainstem

In awake mice, we tested neuronal responses to whisker deflections by targeting silicon probes to different locations in the lateral SC (Fig 2A and 2B), which has been shown in anesthetized animals to be whisker sensitive [26]. Whisker deflections were induced with an airpuff to whiskers contralateral to the recording sites. We found a varying degree of whisker-modulated collicular neurons across 12 recordings from 8 mice with approximately 30% of the units with significant modulation (325/1,005 units modulated, 302/325 positive, 23/325 negative; Figs 2C and S10, respectively). Whisker responses were clearly bimodal, with fast and slow components corresponding to "trigeminotectal" and "corticotectal" SC drive, respectively [25]. The locations of the probes were determined by post hoc localization of dye replacement. To visualize the location of whisker-sensitive units in the LSC, we then pooled and mapped all recorded units onto LSC outlines in coordinate space and computed their modulation indices (Fig 2C).

Having identified the whisker-sensitive region in the LSC, we next wanted to determine the long-range inputs to that region. To do so, we targeted a retrograde virus

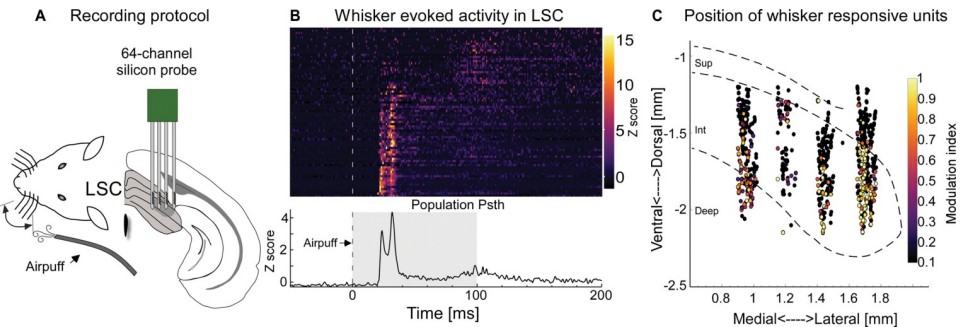

**Fig 2. Whisker sensitivity in the lateral LSC.** (**A**) Experimental schematic of whisker airpuff stimulation and silicon probe recording in LSC in awake mice. (**B**) Top: Single unit PSTHs (rows) in response to whisker stimulation (stippled line) from one example awake recording; units were ordered by first-spike latency, with shortest latencies on the bottom (78 units, 1 mouse). (**C**) Summary from 12 recordings (1,005 units, 8 mice) mapped onto SC outlines through trilateration in CellExplorer [27,28]. Each dot depicts the location of a unit; colors indicate the positive modulation strength upon whisker stimulation (see Materials and methods). The small proportion of negatively modulated units is shown in S10 Fig. The data for Fig 2B and 2C can be found at: https://doi.org/10.11588/data/DNOSZG.

(rAAV-ChR2-tdTomato; [29,30], Fig 3A) to the whisker-sensitive region in LSC, based on stereotaxic coordinates estimated from the recordings (Fig 2). LSC-projecting neurons were found in multiple cortical and subcortical areas (Fig 3B), including whisker-related areas in the ipsilateral whisker motor cortex (MC), in the barrel cortex (BC), and in the contralateral trigeminal complex in the brainstem (Bs), (Fig 3C–3E; whisker motor cortex [31] MC: M1 and M2; see S1 Fig). In addition to these whisker-related projections to SC, we found SC-projecting neurons in the auditory, insular, and ectorhinal cortices as described before [16].

The laminar profiles of LSC-projecting neurons in MC and BC illustrate that in both cortices, cortico-collicular neurons originate in layer 5 (L5) (Fig 3C and 3D). The L5 origin of cortico-collicular projections was further confirmed by registering LSC-projecting somata with layer 6 (L6)-specific EYFP fluorescence in the Ntsr1-EYFP reporter mouse line, showing segregation of these 2 subcortical projection neuron types (Fig 3C). Moreover, virus-mediated labeling of cortical boutons revealed dense innervation of LSC by MC and BC, with moderately sized cortico-collicular boutons (median diameters: BC 1.15 μm, MC 1.38 μm; S2 Fig), consistent with recent reports [32,33]. In summary, the whisker-sensitive region of the LSC receives direct and dense input from whisker areas in the brainstem and from L5 neurons in the barrel and motor cortices.

## Cellular organization of somatosensory and motor pathways in the lateral SC

We next sought to directly target MC-, BC-, and Bs-recipient neurons (RNs) in the LSC to study their organization, cellular identity, and projection patterns. To do so, we employed a *trans*-synaptic anterograde approach, which is based on the ability of the AAV1 serotype to *trans*-synaptically jump to postsynaptic neurons [4]. We injected AAV1-Cre in combination with AAV2-DIO-mCherry into MC, BC, and Bs to simultaneously label the injection sites and their axonal projections with mCherry (red) and to express cre in the synaptically connected postsynaptic target neurons in LSC. Finally, Cre-expressing RNs were revealed by injecting AAV2-DIO-EGFP into LSC (green) (Figs 4A, 4B, S3 and S9). This strategy allowed us to visualize and reconstruct the 3 RN populations (MC-RNs, 6 mice; BC-RNs, 6 mice; Bs-RNs, 7 mice) and register them to standard anatomical borders within SC. This revealed that the 3 whisker-related input pathways predominantly target neurons in the intermediate layer of the

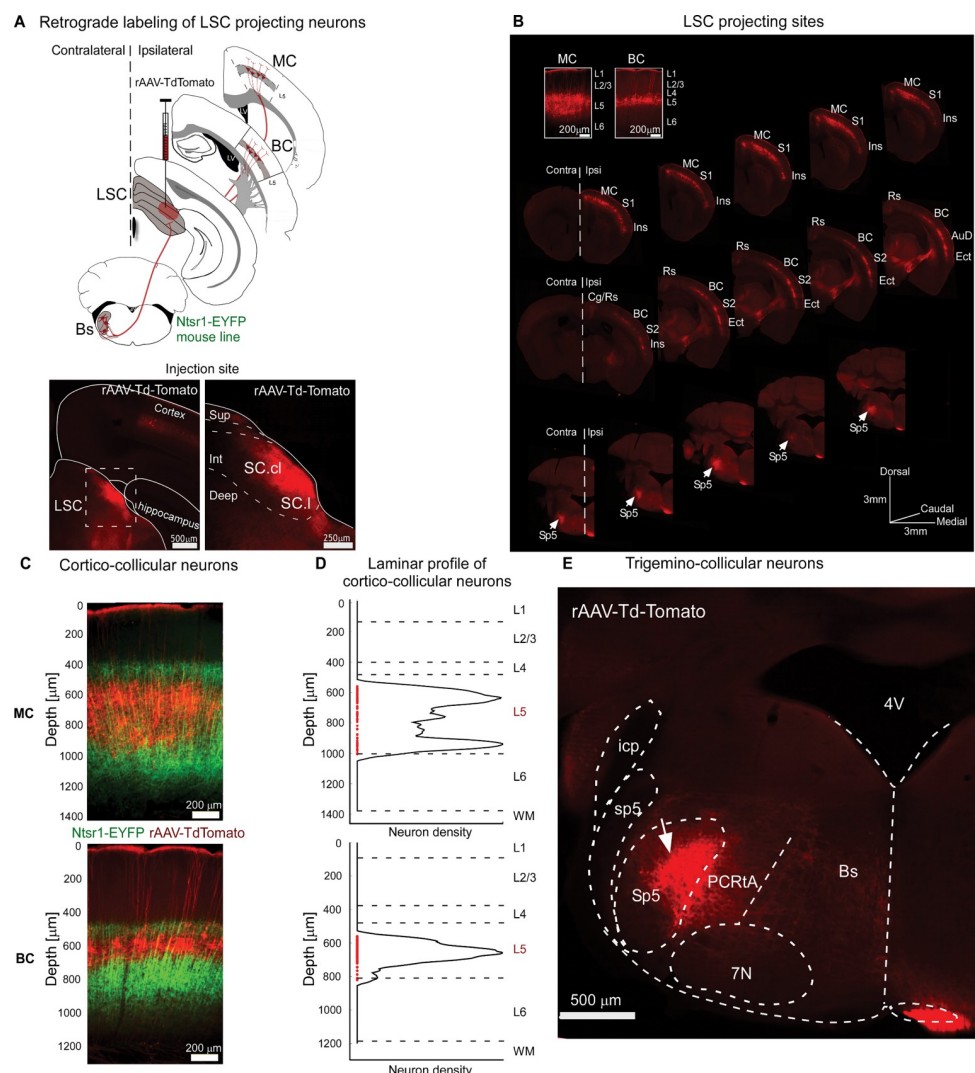

**Fig 3. The whisker-sensitive LSC receives projections from whisker-related cortical and brainstem regions.** (**A**) Upper: Retrograde viral labeling of LSC-projecting neurons: rAAV-tdTomato (red) injection into LSC (Ntsr1-ChR2-EYFP mouse line). Lower: Example images of injection site in LSC (red). (**B**) Consecutive fluorescent images (rostral to caudal) of labeled LSC-projection areas, including ipsilateral MC and BC, and contralateral Sp5 in the trigeminal complex. Insets show L5 neurons in MC and BC at higher magnification. (**C**) Fluorescence images of MC and BC (coronal slices) showing LSC-projecting neurons in layer 5 (red, TdTomato) relative to genetically identified layer 6 neurons (green, Ntsr1-EYFP line). (**D**) Depth and laminar distributions of LSC-projecting neurons in MC and BC. Red dots depict the soma depths of LSC-projecting neurons (relative to pia = 0 µm, MC: median depth −741, IQR = 272 µm; BC: median depth = −648, IQR = 71 µm). Depth-resolved soma densities (black solid lines) are shown relative to layer borders and WM (dashed horizontal lines, estimated based on DAPI signals). Exact N in S2 Table. (**E**) LSC-projecting neurons (red, Td-Tomato) in the contralateral brainstem are located in the Sp5. The data for Fig 3D can be found at: https://doi.org/10.11588/data/DNOSZG. AuD, auditory cortex; BC, barrel cortex; Bs, brainstem; Cg, cingulate cortex; Ect, ectorhinal cortex; icp, inferior cerebellar peduncle; Ins, insular cortex; IQR, interquartile range; MC, motor cortex; PCRtA, parvocelullar reticular nucleus alpha portion; Rs, retrosplenial cortex; S1, primary somatosensory cortex; S2, secondary somatosensory cortex; sp5, spinal trigeminal tract; Sp5, spinal trigeminal nucleus; WM, white matter; 4V, fourth ventricle; 7N, facial nucleus.

LSC (Fig 4C). The RN types largely overlap along the antero-posterior axis (Fig 4D) but are more segregated in the dorsal-ventral and medial-lateral dimensions. Here, the organization shows a lateral Bs-recipient zone, adjacent to a medial cortical-recipient zone with considerable overlap between MC- and BC-RNs (Fig 4E and 4F). We overlaid these recipient zones on

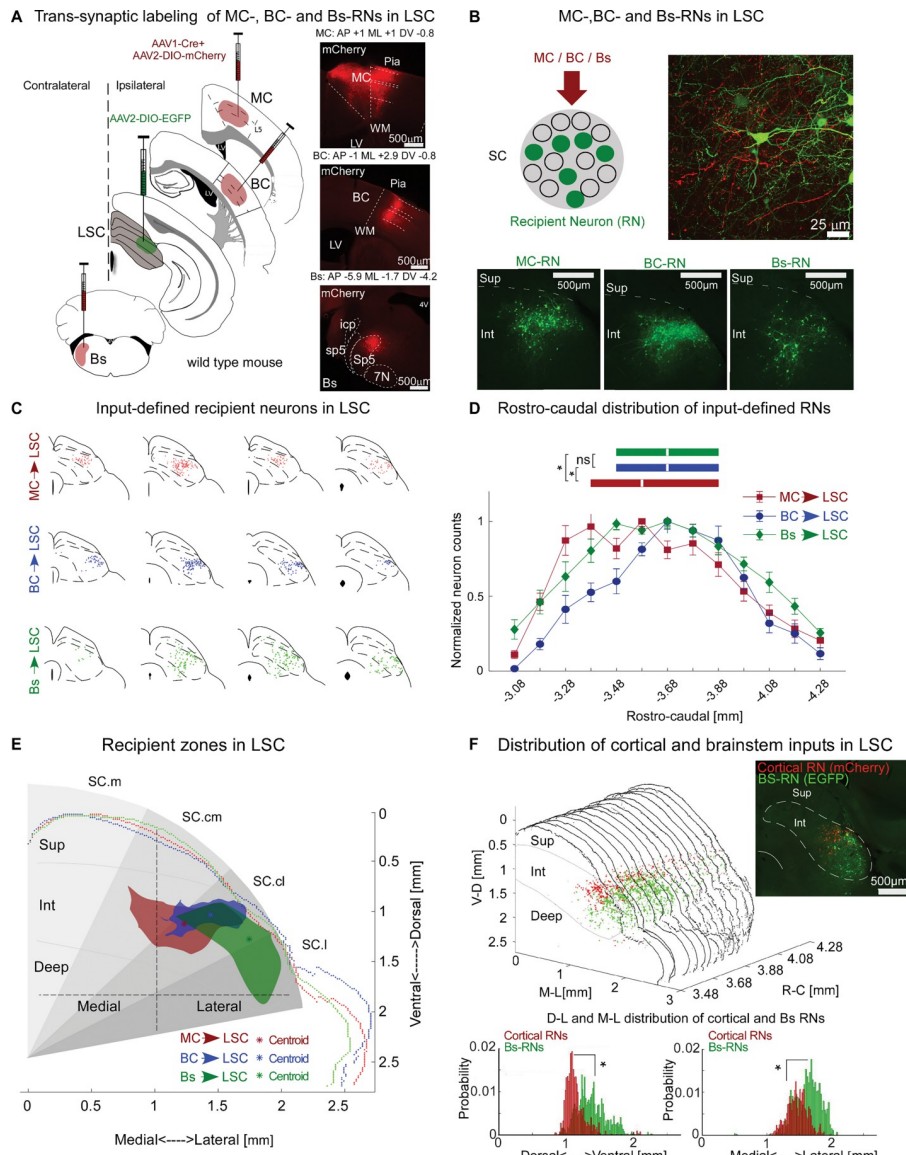

**Fig 4. Whisker-related sensory-motor RNs are organized into overlapping zones within the intermediate layer the LSC.** (**A**) *Trans*-synaptic labeling of MC-, BC-, and Bs-RNs in SC (left). Cocktails of AAV1-Cre + AAV-DIO-mCherry were injected into projecting sites (MC, BC, or Bs) and AAV2-DIO-EGFP in SC. Example fluorescent pictures (right) of mCherry expression (red) in the projecting sites. (**B**) Top left: Schematic of *trans*-synaptically labeled RNs (green). Top right: Example confocal image of SC showing mCherry-expressing BC axons (red) and Cre-dependent expression of EGFP in BC-RNs (green). Bottom: Fluorescent images of EGFP expression (green) in MC-, BC-, and Bs- RNs. (**C**) Example reconstructions of the 3 RN populations along the rostro-caudal axis (red: MC-RNs; blue: BC-RNs; green: Bs-RNs), registered to standard anatomical borders within SC. (**D**) Anterior–posterior distributions of RNs. Data points show mean RN counts per 100 μm bin (red: MC-RNs, 6 mice; blue: BC-RNs, 6 mice; green: Bs-RNs, 7 mice) normalized to their maximum count. Box plots show medians (line in box) and IQRs (first to third quartile) in mm (boxes), ([median, Q1, Q3 in mm] MC-RNs: 3.58, 3.38, 3.88; BC-RNs: 3.68, 3.48, 3.88; Bs-RNs: 3.68, 3.48, 3.88). (**E**) Comparison of recipient zones in the DV and ML dimensions in SC. Fluorescent thresholded representative slices of each RN population, registered at a similar AP coordinate and to approximate radial SC zones according to [16]. (Centroids: [DV, ML, μm] MC: 1,123, 1,505; BC: 1,035, 1,297; Bs: 1,276, 993). (**F**) Colabeling experiment of cortical (red, MC-RNs + BC-RNs) and Bs-RNs (green). Top right: Example fluorescent image showing cortical and peripheral RNs. Middle: Fluorescence thresholded RN signals from 15 consecutive images (1 brain). Bottom: Histograms show the thresholded pixel gray value probability for cortical-RNs and Bs-RNs in the DV axis and in the ML axis. ([DV, median and IQR, relative to SC dorsal surface] Cortical-RN [μm]: 1,115, 147; Bs-RN: 1,365, 281; [ML, median and IQR, μm relative to SC midline] Cortical-RN: 1,472, 211; Bs-RN: 1,637, 287, *p* < 0.001). * represents *p* < 0.01; D: Kruskal–Wallis, F: Wilcoxon rank-sum; exact *p*-values in S1 Table, exact N in S2 Table. Data are shown as

mean ± SEM. The data for Fig 4D–4F can be found at: https://doi.org/10.11588/data/DNOSZG. BC, barrel cortex; Bs, brainstem; DV, dorsal-ventral; icp, inferior cerebellar peduncle; IQR, interquartile range; LSC, lateral SC; LV, lateral ventricle; MC, motor cortex; ML, medial-lateral; Pia, pia mater; RN, recipient neuron; SC.m, medial superior colliculus; SC.cm, superior colliculus centromedial; SC.cl, centrolateral superior colliculus; SC.l, lateral superior colliculus; sp5, spinal trigeminal tract; Sp5, spinal trigeminal nucleus; WM, white matter; 7N, facial nucleus.

the approximate outlines of the radial zones determined by Benavidez and colleagues [16]. The comparison shows that recipient zones fall into the radial zones as follows: BS → lateral / centrolateral; BC → centrolateral; MC → centrolateral / centromedial (Fig 4E).

Together, the anterograde and retrograde tracing experiments identify a region within the intermediate layers of the LSC, which spans approximately 1.2 mm in the rostro-caudal axis (approximately 85% of SC's extent) and which receives input from main whisker circuits in the cortex and brainstem. This region is highly whisker sensitive (Fig 2), suggesting that the intermediate layer of the LSC is a "whisker SC" located ventrally to the "visual SC."

## Cortical and brainstem pathways directly target GABAergic neurons in the LSC

Are the neurons targeted by cortical and brainstem long-range input to the LSC excitatory or inhibitory? To address this question, we first estimated the proportion of GABAergic neurons in the LSC. Using a GAD-GFP mouse, in which GABAergic neurons express GFP, we colabeled all neurons using the pan-neuronal marker NeuN-Alexa 647 (Fig 5A) and estimate that approximately 23% of the LSC neurons in the intermediate layer are GABAergic (Figs 5F and S6), which is slightly lower than in the superficial SC (approximately 30%; [15]).

We then tested for monosynaptic innervation of GABAergic LSC neurons by employing a *trans*-synaptic intersectional strategy that allowed us to separate GABAergic RNs (iRNs) from the population of RNs, for each long-range input pathway (Fig 5B). In GAD-Cre mice, which express Cre in GABAergic neurons [34], we injected anterograde virus (AAV1-Flpo) into MC, BC, or Bs, as well as conditional reporter viruses (AAV-ConFon EYFP, AAV-fDIO mCherry) into LSC. This intersectional approach differentially labeled RNs with mCherry and iRNs with EYFP, demonstrating direct innervation of GABAergic neurons by MC, BC, and Bs input pathways (Figs 5C and S4 and S5 for controls). Reconstructions of the 2 populations show that iRNs and RNs are intermingled for all 3 input pathways (Fig 5D). Our estimates of the iRN/RN proportions suggest that each pathway targets between 34% and 37% GABAergic neurons along the full extent of the recipient antero-posterior axis (Figs 5E–5G and S11).

In summary, long-range input pathways from MC, BC, and Bs extensively target GABAergic neurons in the LSC. The proportion of iRNs exceeded the proportion of GABAergic neurons (>34% versus 23%; Fig 5G), suggesting a preferential targeting for GABAergic neurons by these long-range input pathways.

## Convergence of somatosensory and motor cortex in LSC neurons

The considerable overlap between recipient zones in LSC observed in Fig 4 suggests possible convergence of long-range input pathways on the level of individual LSC neurons. To test for monosynaptic input convergence in single LSC neurons (CVG-RN), we employed an intersectional strategy as follows: In the same mouse, we injected 2 projection sites—for example, MC and BC—each with a different *trans*-synaptic variant—AAV1-Cre or AAV1-Flpo. Injections of the double-conditional (Cre- and Flpo-dependent) reporter virus in SC indeed revealed CVG-RNs for all 3 pathways (Fig 6A and 6B). MC and Bs convergence is highest in the anterior LSC, while MC and BC and BC and Bs convergences peak in the center and posterior SC, respectively (Fig 6C).

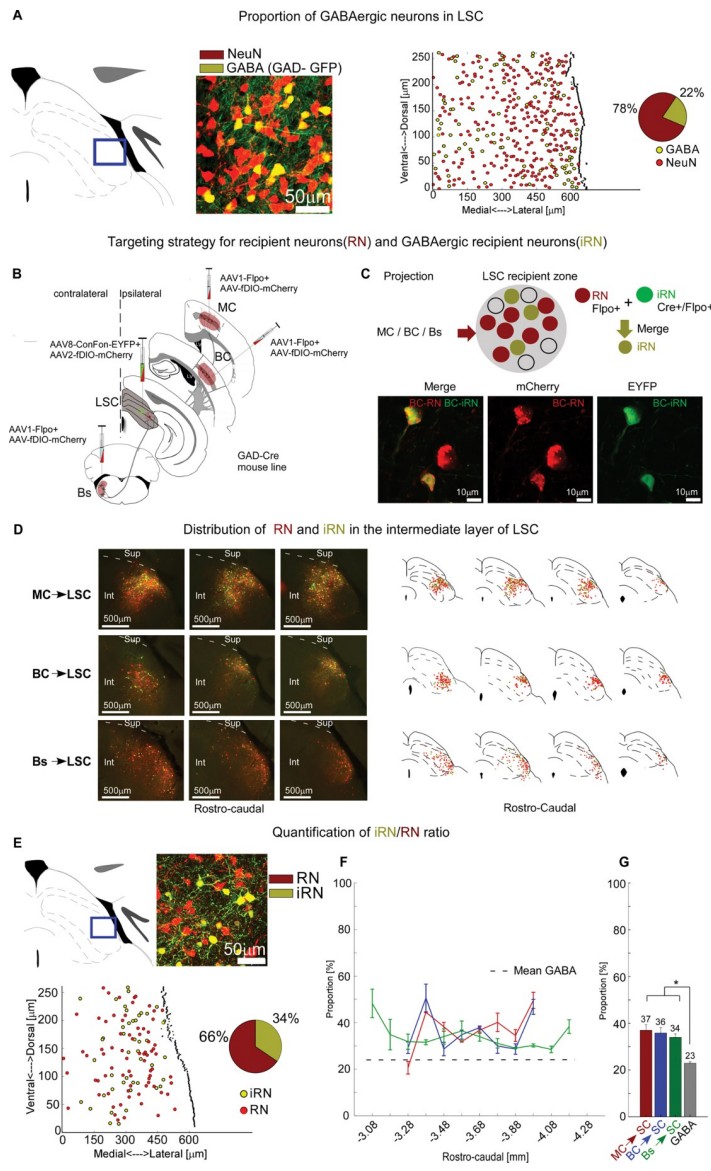

**Fig 5. Cortical and brainstem pathways directly target GABAergic neurons in LSC.** (**A**) The proportion of GABAergic neurons in the LSC was determined in a GAD-GFP mouse by calculating the proportion of GFP-expressing neurons (GABA, yellow) and all neurons labeled with pan-neuronal marker NeuN-Alexa 647 (red). From left to right: Example single rostro-caudal section; corresponding fluorescence image of GFP and NeuN signals; reconstruction of soma locations; pie chart of GABA/NeuN proportions for this section (84 GABA/384 NeuNs, 21.9%). The proportion for all analyzed sections ($n$ = 30, 10 slices, 1 brain) was 23.1 ± 1.2% (see also S6 Fig). (**B**) Intersectional labeling of RNs and iRNs for each pathway in GAD-cre mice. AAV1-Flpo + AAV-fDIO-mCherry was injected in the projecting sites (MC, BC, and Bs), and AAV8-Con/Fon-EYFP + AAV2-fDIO-mCherry in LSC of GAD-Cre mice to label iRNs and RNs, respectively. (**C**) Top: Schematic of *trans*-synaptically labeled RNs (red) and iRNs (green). Bottom: Confocal images of BC-iRNs and BC-RNs. (**D**) Distribution of iRNs and RNs. Left: Example fluorescence images of iRNs and RNs in different SC sections for MC, BC, and Bs pathways, respectively. Right: Example reconstructions of MC-, BC-, and Bs-iRN and RN populations along the rostro-caudal axis (red: RNs, yellow: iRNs), registered to standard anatomical borders within SC. (**E**) Example for quantification of iRN/RN ratio for the MC-SC pathway. Fluorescence image, reconstruction, and pie chart summary of iRNs/RNs proportion (41 iRNs/119 RNs, 34%). (**F**) Summary of iRN/RN proportion for all 3 input pathways (MC red, BC blue, Bs green) along the rostro-caudal SC axis and GABA/NeuN proportion (grey dashed line). (**G**) Means and SEMs of iRN/RN proportions and GABA/NeuN for all 3 input pathways (same colors as in F); [mean per slice ± SEM, iRN/RN or GABA/NeuN, n mice]; MC: 36.4 ± 2.4%, 2; BC: 35.6 ± 2.4%, 3; Bs: 33.2 ± 1.5%, 3; GABA/NeuN: 23.1% ± 1.2, 1). * represents $p < 0.01$; Kruskal–Wallis; exact $p$-values in S1 Table, exact N values in S2 Table. Data are shown as mean ± SEM. The data for Fig 5A, 5E and 5F can be found at: https://doi.org/10.11588/data/DNOSZG. BC, barrel cortex; Bs, brainstem; Int, Intermediate layers; LSC, lateral SC; MC, motor cortex; RN, recipient neuron; Sup, Superficial layers.

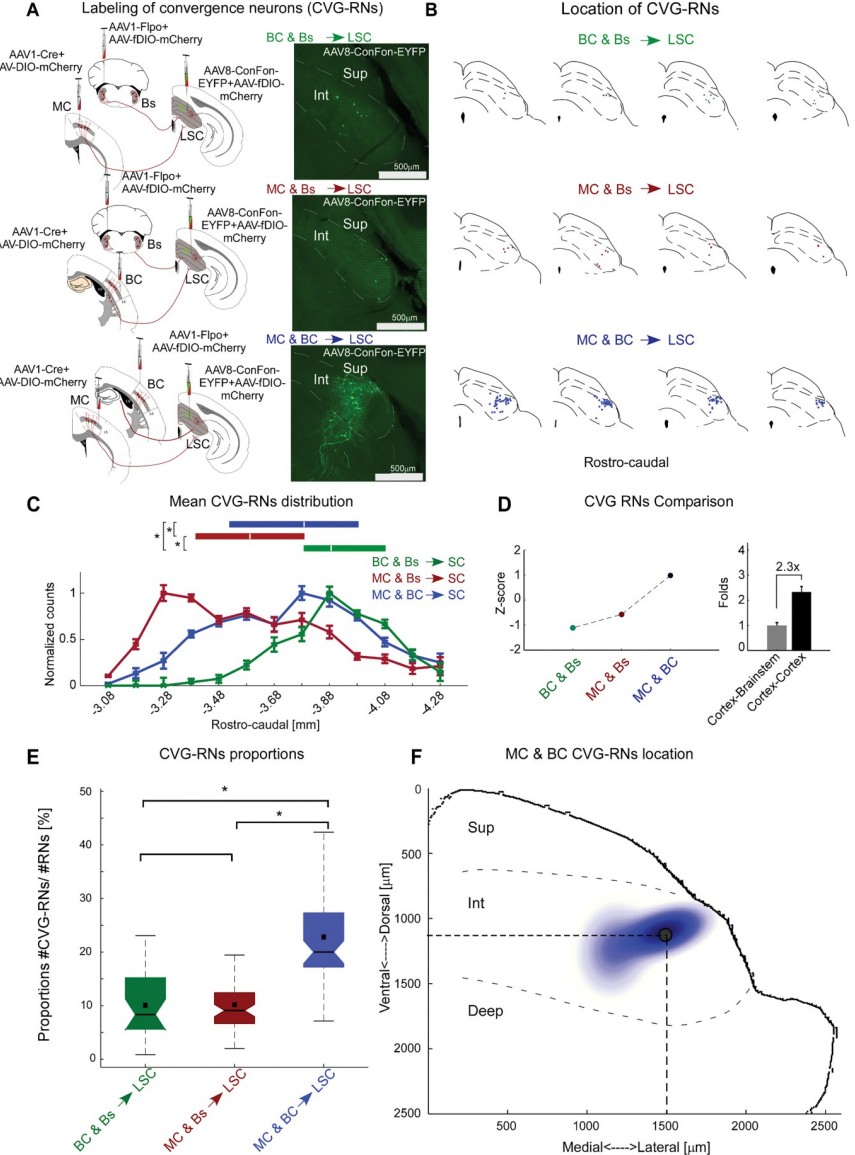

**Fig 6. Whisker-related input pathways converge in subpopulations of LSC neurons.** (**A**) Left: Targeting convergence neurons (CVG-RNs) in LSC. AAV1-Cre + AAV2-DIO-mCherry and AAV1-Flpo + fDIO-mCherry were injected into 2 projecting sites and AAV8-Con/Fon-EYFP + AAV2-DIO-mCherry into LSC to reveal CVG-RNs and RNs, respectively. Right: Example fluorescent images of CVG-RNs. (**B**) Example reconstructions of CVG-RNs along the rostro-caudal axis, registered to standard anatomical borders within SC. (**C**) Normalized distributions of CVG-RNs along the rostro-caudal axis (bin size 100 µm). Box plots above show medians (white lines), and IQR (first to third quartile, boxes). MC and BC, MC and Bs, BC and Bs *n* = 4, 3, 2 mice, respectively. (**D**) Left: Z-scored mean number of CVG-RNs for each input pair. Right: Fold difference between mean number of CVG-RNs per 100 µm bin for cortex-cortex (4 mice) and brainstem-cortex (5 mice). (**E**) Proportions of CVG-RNs to RNs for each convergence input pair. Box plots show the median (line in box), mean (square in box), and IQR (first to third quartile, boxes). ([median, mean, IQR] MC and BC: 0.20, 0.22, 0.01; BC and Bs: 0.08, 0.01, 0.09; MC and Bs: 0.09, 0.01, 0.06). (**F**) The two-component Gaussian Mixture Model shows the area of MC and BC convergence zone in the intermediate layer of the LSC (11 slices, 2 mice, 222 neurons). The black dot indicates the location of the highest probability of finding MC and BC CVG-RNs (1,126 µm from SC surface and 1,492 µm from the midline). * represents *p* < 0.01; C, E: Kruskal–Wallis; exact *p*-values in S1 Table, exact N in S2 Table. Data are shown as mean ± SEM. The data for Fig 6C–6F can be found at: https://doi.org/10.11588/data/DNOSZG. BC, barrel cortex; Bs, brainstem; Int, intermediate layer; IQR, interquartile range; LSC, lateral SC; MC, motor cortex; RN, recipient neuron; SC, superior colliculus; Sup, superficial layer.

To quantify the degree of convergence, we compared (1) z-scored mean CVG-RN counts per input pair and (2) the proportion of CVG-RNs with respect to RNs between all input pairs. Both analyses revealed that MC and BC convergence stands out compared to the other convergence pairs. Notably, the z-scored MC and BC CVG-RN estimates a 2.33-fold higher deviation compared to Bs and cortex convergence (Fig 6D).

To estimate the proportion of convergence neurons relative to all recipient neurons, we counted mCherry-labeled RNs and EYFP-labeled CVG-RNs (Figs 6A and S12). Cortico-cortical convergence was estimated to be about 2.3-fold higher than brainstem-cortical convergence (approximately 23% CVG-RN for the MC and BC input pair versus approximately 10% CVG-RN for both the BC and Bs and MC and Bs input pairs) (Fig 6E). Fitting a two-component Gaussian Mixture Model to the coordinates of MC and BC CVG-RNs localized the cortico-cortical convergence zone to the dorsal portion of the intermediate layer (Fig 6F).

Thus, motor and barrel cortex pathways converge on single neurons in the LSC, highlighting the whisker SC as a node for the integration of somatosensory and motor cortical signals. This convergence motif could potentially support fast, temporally precise integration of signals from different cortical regions; therefore, we next tested this possibility at the functional level.

## Single LSC units integrate sensory- and motor cortex inputs

To validate functional convergence of MC and BC in LSC, we made electrophysiological recordings in LSC in combination with optogenetic stimulation of L5 neurons in MC and BC in anesthetized mice. We used Rbp4-Cre-ChR2-EYFP mice (*n* = 4), in which L5 neurons express ChR2 [35,36] and successively stimulated MC-L5 and BC-L5 during the same experiment by using a movable fiber optic placed above the pial surface (Fig 7A).

By characterizing the spiking activity of individual SC units in response to successive optogenetic stimulation of L5 in MC or BC (Fig 7B), we identified 30 units that responded to either MC-L5 or BC-L5, from which 9 responded to both (Fig 7C). Unit spike latencies in response to optogenetic stimulation of L5 were approximately 9 to 12 ms (Fig 7D, exact values in S3 Table), comparable to the latencies of optogenetically evoked spiking in L5 and L5-innervated downstream neurons in the thalamus [37,38], suggesting monosynaptic activation of LSC neurons via L5-SC synapses.

Thus, together with the anatomical tracing, these results show that the LSC integrates dual functional input from both motor and barrel cortex layer 5 neurons in a subpopulation of about 20% to 30% of the recipient neurons.

## *Trans*-collicular pathways via cortico- and trigemino-collicular projections to the diencephalon and brainstem

It is not known whether cortico-collicular signals can be routed directly to LSC downstream targets via monosynaptic *trans*-collicular pathways. If this is the case, cortico-collicular input neurons in the LSC would be required to form long-range pathways that leave the SC. To test this possibility, we leveraged the *trans*-synaptic labeling approach to search for axonal signals from RN labeling experiments (3 mice) throughout the brain. Indeed, we observed abundant RN axons and varicosities in multiple target areas in the brainstem and diencephalon (reconstructions in Fig 8A; high magnification examples in S7 Fig).

To estimate relative innervation strengths of the target nuclei, we computed RN-output maps from reconstructed axon counts (Fig 8B) to visualize relatively enriched or sparse innervation. All 3 *trans*-collicular pathways innervated multiple diencephalic nuclei, with the highest axonal density in ZI and sensory-motor–related nuclei in the thalamus, i.e., the posterior (PO), ventro-medial (VM) and parafascicular (PF) nuclei. In the brainstem, the strongest

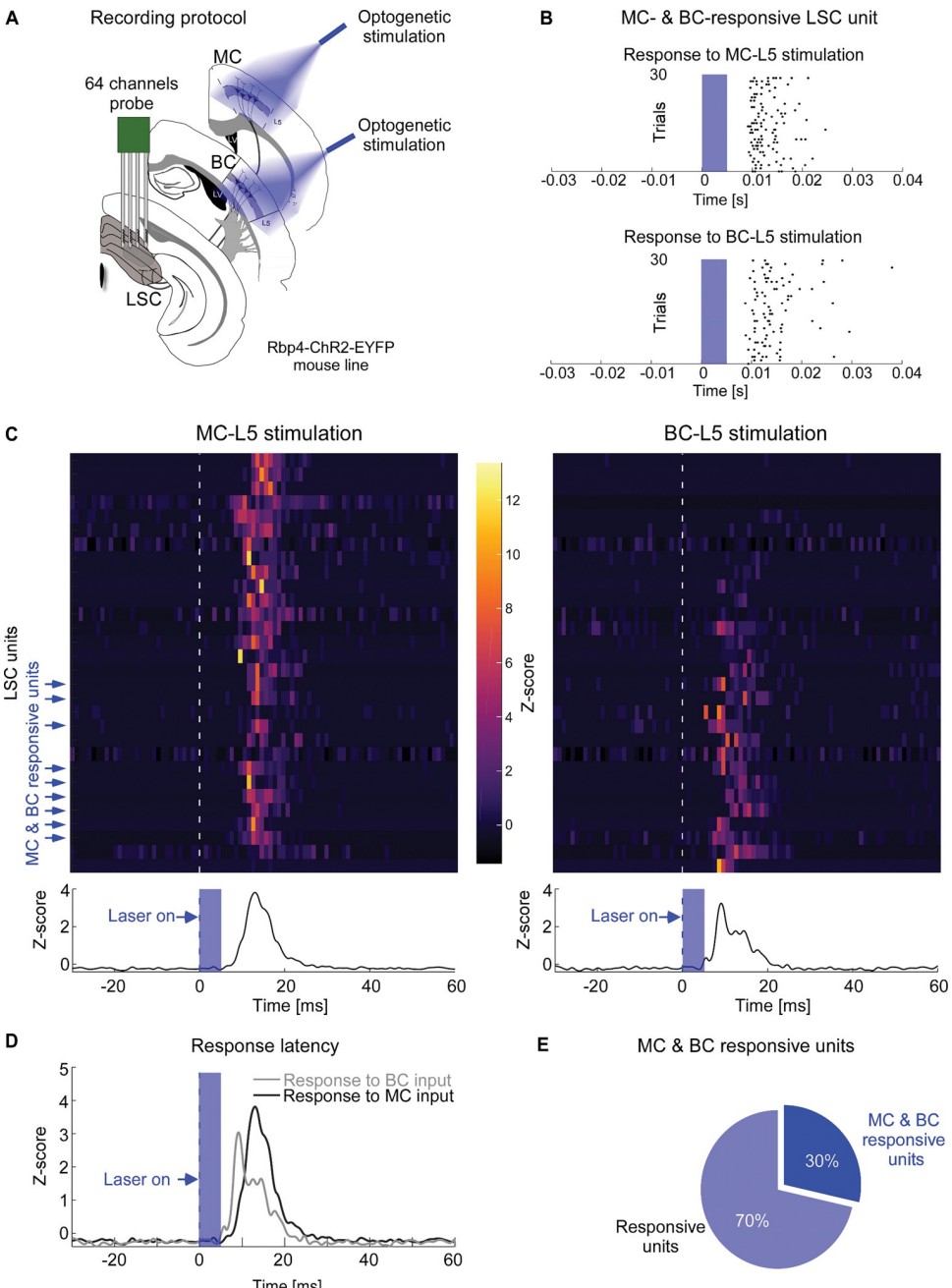

**Fig 7. MC and BC functional convergence in LSC.** (**A**) Schematic of the optogenetic stimulation of MC-L5 and BC-L5 and 64-channel silicon probe recording in LSC of anesthetized Rbp4-Cre-ChR2-EYFP mice. (**B**) Example raster plot of a single unit responding to both MC-L5 (top) and BC-L5 (bottom) laser stimulation (blue, 30 trials). (**C**) Top: Single unit PSTHs in response to MC-L5 (left) and BC-L5 (right) laser stimulation (dashed line). Units were ordered by response magnitude upon BC stimulation. Among 30 responsive units (out of 409 recorded SC units), 25 were responsive to optogenetic stimulation of MC-L5, 14 to BC-L5, and 9 to both MC-L5 and BC-L5 (blue arrows). Bottom: LSC population PSTHs for MC-L5 (left) and BC-L5 (right) laser stimulation (blue, laser on). (**D**) Overlaid LSC population PSTHs for MC-L5 (black) and BC-L5 (gray) laser stimulation (blue). (**E**) Proportions of MC-L5 or BC-L5 responsive units (blue) and units responding to both (dark blue). See S3 Table for extended data on unit population spiking characteristics. The data for Fig 7B–7E can be found at: https://doi.org/10.11588/data/DNOSZG. BC, barrel cortex; LSC, lateral SC; MC, motor cortex.

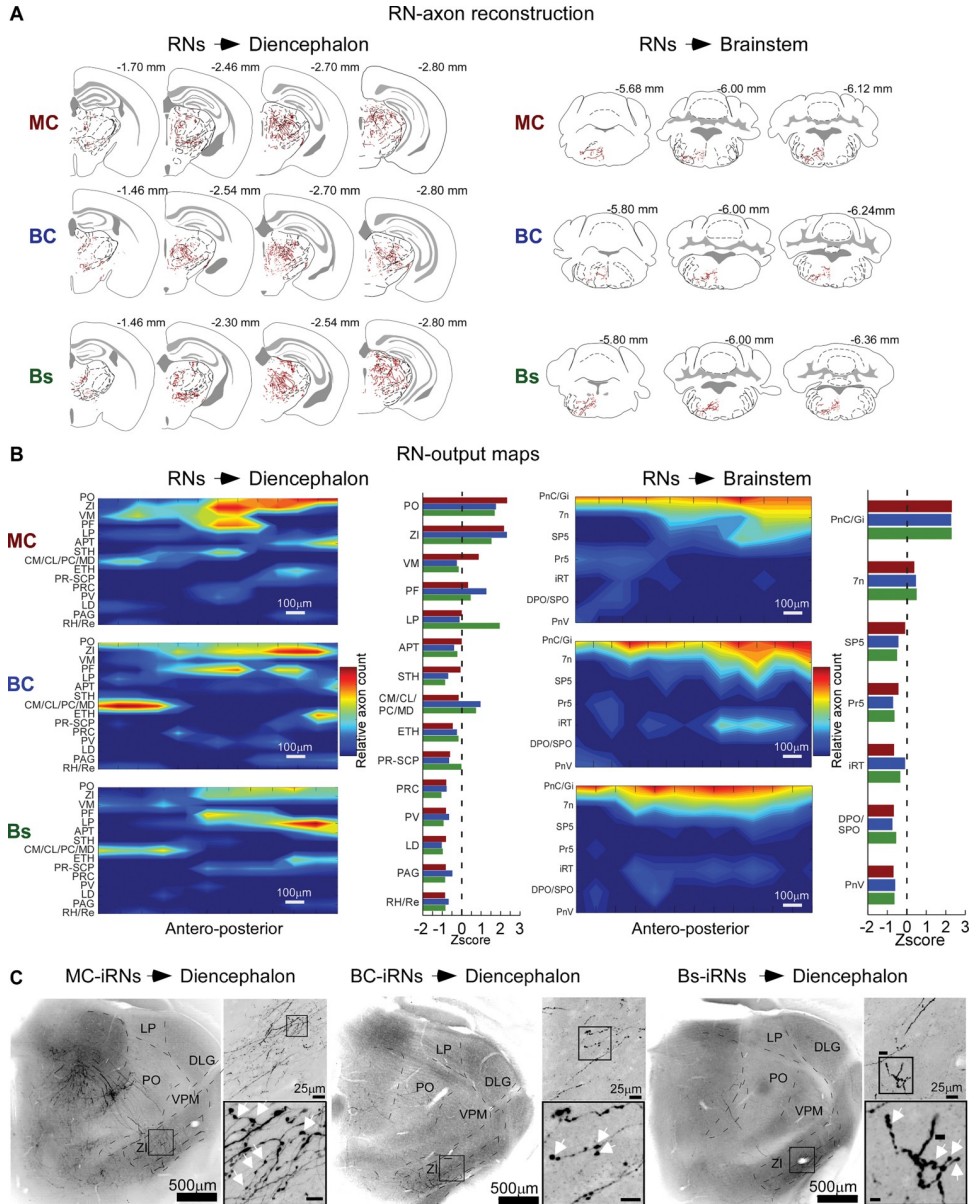

**Fig 8. *Trans*-collicular pathways via cortico- and trigemino-collicular projections to the diencephalon and brainstem.** (**A**) Example axon reconstructions of MC-, BC-, and Bs-RNs in diencephalic (left) and brainstem (right) target nuclei. (**B**) RN output maps computed from normalized RACs along the rostro-caudal axis for all 3 RN pathways and their diencephalic (left) and brainstem (right) target nuclei. Red and blue indicate high or low RAC values, respectively. Bar plots summarize Z-scored axon counts for each target nucleus, normalized to the maximum axon count in each pathway. Stippled line indicates Z-score = 0, $n$ = 3 mice; see Materials and methods for details. (**C**) Example confocal images (desaturated and inverted) of axons from MC-, BC-, and Bs-iRN in the diencephalon. Inset images at a higher magnification show varicosities (white arrowheads, scale bar 5 μm). See S7 Fig for higher magnification images. The data for Fig 8B can be found at: https://doi.org/10.11588/data/DNOSZG. APT, anterior pretectal nucleus; CL, central lateral nucleus; CM, central medial nucleus; DLG, dorsolateral geniculate nucleus; ETH, ethmoidal nucleus; LD, laterodorsal nucleus; LP, lateral posterior nucleus; MD, mediodorsal nucleus; PAG, periaqueductal gray; PC, paracentral nucleus; PF, parafascicular nucleus; PO, posterior nucleus; PR, prerubral field; PRC, precommissural nucleus; PV, paraventricular nucleus; RAC, relative axon count; Re, Reuniens nucleus; RH, rhomboid nucleus; SCP, superior cerebellar peduncle; STH, subthalamic nucleus; VM, ventromedial nucleus; VPM, ventral posteromedial nucleus; ZI, zona incerta.

innervation was found in the pontine reticular (PnC; caudal part), the gigantocellular reticular (Gi), and the facial nucleus (7N). The innervation strength of a given target was often similar for the 3 *trans*-collicular pathways, particularly for brainstem targets. The output maps differed more strongly in the thalamus, for example, in the lateral-posterior nucleus (LP), which received relatively enriched innervation by Bs-RN axons.

We next analyzed the inhibitory RN pathways and asked if they solely resemble classic interneurons, which only project within intracollicular circuits, or if they also give rise to long-range projections outside the SC (GABAergic projection neurons). Unexpectedly, we found that iRNs from all 3 pathways give rise to high density innervation of the ZI as well as projections to PO and PF (Figs 8C and S8). In contrast, no iRN projections were found in the brainstem. Thus, *trans*-collicular pathways extend to long-range GABAergic output with hitherto unknown inhibitory functions in thalamic nuclei and the ZI. Based on the observed axonal projections, both RNs and iRNs innervate the dorsal and ventral ZI (Figs 8A, 8C, S7A, S7B, S8A and S8B). Since the ZI is a main source of GABAergic projections to higher-order thalamus [39], this connectivity motif suggests inhibitory and disinhibitory gate control of higher-order thalamus via RN and iRN pathways, respectively.

In summary, we found excitatory and inhibitory monosynaptic *trans*-collicular pathways (Fig 9), suggesting that at least part of the input signals from cortex and brainstem are directly transformed into SC output to rapidly route cortical and brainstem information to diverse points along the neural axis.

## Discussion

### *Trans*-collicular pathways

The superior colliculus—a highly conserved sensory-motor structure in vertebrates—is said to be one of the best-characterized structures in the brain [2,14,15]. However, precise understanding of how particular input pathways to the SC are synaptically connected to particular SC output targets has been hampered due to the lack of tools to determine the connectivity features of input-defined SC neurons. Moreover, such anatomical constraints are necessary to develop functional predictions and testable hypotheses for sensory-motor computations in the SC and, in a broader context, to understand how cortex-mediated and SC-mediated sensory-motor arcs relate to each other.

The present study elucidated the input–output organization of the mouse LSC by tracing input-defined pathways from the brainstem and motor and somatosensory cortices to the LSC and further on to collicular downstream targets in the diencephalon and brainstem (Fig 9). We thereby identified a connectivity scheme with significant implications for collicular computations: *trans*-collicular pathways, monosynaptically traversing through the LSC and directly linking cortical and brainstem whisker circuits to downstream targets in the brainstem and diencephalon. The connectivity scheme followed by these *trans*-collicular neurons differs from that found in other layered circuits—for example, compared to the canonical circuit framework of the cortex, in which layer 4 neurons are considered input neurons that form only intracortical connections. In contrast, the SC input neurons identified here directly provide long-range output axons to multiple subcortical targets, suggesting that at least part of the input signals are directly transformed into SC output via *trans*-collicular pathways.

In awake animals, we found that single neurons in the intermediate layer of the LSC are highly sensitive to whisker stimulation, in accordance with earlier work in anesthetized rodents that showed whisker sensitivity in the LSC [26] and trigeminal and cortical input from the somatosensory barrel and motor cortex [16,25]. Our results expand on this work by demonstrating that the whisker-sensitive LSC is densely interconnected with whisker circuits

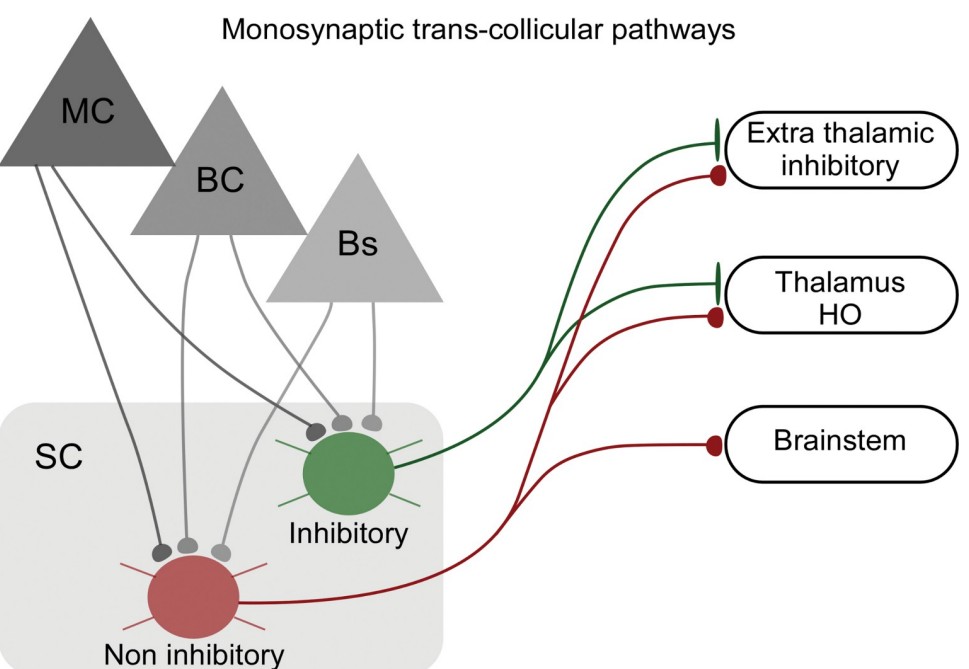

**Fig 9. Summary of monosynaptic *trans*-collicular pathways of the whisker SC.** Motor-sensory long-range inputs innervate inhibitory and noninhibitory projection neurons in the LSC. Noninhibitory pathways target nuclei in the diencephalon and Bs, while inhibitory projection neurons target higher-order thalamus and extrathalamic inhibitory nuclei and not the Bs. Bs, brainstem; LSC, lateral SC; SC, superior colliculus.

throughout the brain, both in terms of inputs and outputs, together suggesting that the intermediate layer of the LSC serves as a whisker SC.

The microanatomical analyses and mapping data we present here allow several functional predictions for whisking behavior. Firstly, we found that brainstem long-range inputs to the LSC turn into direct excitatory outputs to the 7N, a brainstem motor nucleus for whisker movements [40]. This finding predicts a short sensory-motor loop that could mediate the "minimal impingement" phenomenon, which animals use to optimize tactile sensitivity by minimizing the force of tactile sensors upon surface contact [41]. Our data suggest that whisker touch signals ascending from the brainstem trigeminal nucleus to the LSC are monosynaptically transformed into motor output signals to the 7N, as a direct sensory-motor loop to stabilize whisker movement force, while animals move relative to objects.

Furthermore, in the thalamus, the 3 investigated *trans*-collicular pathways from MC, BC, and Bs target higher-order thalamic nuclei, such as PO, but not first-order thalamic nuclei (Fig 8). This finding is particularly relevant for higher-order thalamic computations of cortical signals because it suggests parallel entry routes of cortical L5 signals to the higher-order thalamus: an indirect L5 *trans*-collicular pathway and a direct cortico-thalamic pathway [33,42–44].

Another significant outcome of our mapping data is the link between the whisker and the visual system: whisker trigemino-collicular pathways send abundant projections to the visual LP—the rodent pulvinar in the thalamus (Fig 8). This result provides an anatomical substrate for the recent finding that the LP integrates visual and tactile information [45]. The integration of *trans*-collicular whisker signals in the LP may be an important mechanism underlying visuo-tactile functions such as hunting behavior, in which the SC has been shown to play a key role [8].

## Long-range input convergence in the LSC

The multimodal nature of the SC is well established and functional observations of multisensory SC neurons [18] consistently support multimodal convergence in the SC. However, only few studies directly demonstrated long-range convergence at the level of individual SC neurons using anatomical means [19,46,47], and, to the best of our knowledge, the possibility of specific convergence between motor and sensory cortical pathways in the SC had not yet been tested. Using anatomical and functional tools to test convergence of input pathways from MC, BC, and Bs in individual LSC neurons, we found subpopulations of LSC neurons, receiving monosynaptic input from 2 input pathways. While the estimated proportion of brainstem-cortical convergence neurons was approximately 10%, the proportion of cortico-cortical convergence neurons (MC and BC) was more than twice as high (approximately 23%). We demonstrate that convergence between motor and sensory cortex is functional by showing that approximately 30% of the individual LSC units receiving functional cortical input are responsive to optogenetic stimulation of L5 neurons in both MC and BC.

The integration of convergent L5 signals from sensory and motor cortices by single SC neurons suggests a low-level neuronal substrate for sensory-motor integration in the SC from which functional predictions in the context of the execution and control of motor plans [24] can be derived. In this framework, inputs from MC inform the SC about an intended motor action, i.e., an "efference copy" of the motor command [48], while BC provides information on the resulting sensory status, thereby enabling mechanisms to initiate small motor command readjustments and/or context-dependent dampening or enhancement of sensation, possibly analogous to the SC mechanisms proposed for the stabilization of visual perception [49]. However in the context of this proposition, it also needs to be noted that L5 neurons of both S1 and M1 convey motor-related signals, i.e., are not fully independent [31,50].

## Long-range innervation of GABA neurons in the LSC

The SC contains substantial populations of GABAergic neurons with reported numbers ranging from 30% to 45% [15]. We estimate that the LSC contains approximately 23% GABAergic neurons. Rather unexpectedly, we found that GABAergic neurons are directly innervated by long-range pathways from MC, BC, and Bs. Interestingly, the estimated proportion of innervated GABAergic neurons was between 34% and 37% and thereby significantly exceeded the estimated proportion of GABAergic neurons in the LSC (approximately 23%), suggesting an important functional role for the recruitment of inhibitory SC-circuits by cortical and brainstem long-range inputs. Furthermore, a recent report using anterograde *trans*-synaptic viral tracing found that nearly half of the SC neurons downstream of the secondary motor cortex are GABAergic [24]. Together, these findings suggest that the dual recruitment of inhibitory and excitatory circuits in the SC provides cortico-collicular pathways with the ability to flexibly dampen or enhance SC-mediated innate behaviors.

Our mapping data demonstrate that the populations of targeted GABAergic neurons contain projection neurons. Notably, these GABAergic projections target PO and the ZI but not the brainstem. Thus, the PO and the ZI, the latter of which is a potent source of inhibition for higher-order thalamic nuclei [51], receive both GABAergic and non-GABAergic *trans*-collicular input, therefore predicting bidirectional control of these LSC targets. Future investigations of how the ZI is modulated by whisker signals from parallel GABAergic and non-GABAergic *trans*-collicular pathways will likely generate substantial insight into the role of whisking on ZI's myriad functions, which include locomotion, feeding, hunting, pain regulation, and defense behaviors [7,8].

What is the relation between the "older" collicular and the "newer" cortical arcs of sensory-motor loops? In the mammalian brain, in which both arcs are densely interconnected [52], and as suggested by the present study, one possible distinguishing function is the primary foci of their corresponding "brain-world" loops: self-motion for SC-mediated loops and external objects for cortical-mediated loops. However, it should also be noted that the mammalian SC maintains a direct leg to external sensory signals (for example, the brainstem inputs described here) upon which quick motor actions can most likely be computed independently of the cortex. The cortex may thus provide a dynamic interference conduit to these SC-generated sensory-motor functions to enhance behavioral flexibility—for example, by "vetoing" SC-mediated behaviors via selective recruitment of the GABAergic pathways depicted in Fig 9. Achieving a functional understanding of the interactions between cortical and collicular sensory-motor loops is a valuable challenge for the future.

## Materials and methods

### Ethics statement

All experimental procedures were approved by the governing body (Regierungspräsidium Karlsruhe, Germany, approval numbers: 35–9185.81/G-289/21, T-39-20, 35–9185.82/A-8/20, 35–9185.81/G-216/19) and performed according to their ethical guidelines.

The reagents and resources used in this study are summarized in Table 1.

### Animals

Mice (male and female, 10 to 20 weeks of age at the time of virus injection) were housed with food and water ad libitum on a 12-h light/dark cycle.

### Mouse lines

Retrograde labeling:

"Ntsr1-EYFP"; crossbreed between "Ntsr1-Cre" x "Ai32" (B6.FVB(Cg)-Tg/(Ntsr1-Cre) GN220Sat/Mmucd x B6.129S-Gt(ROSA)26Sortm32(CAG-COP4*H134R/EYFP).

RN labeling: "Wild type"; C57BL/6N.

iRN labeling: "GAD-Cre"; B6-Gad1tm2(Cre)Mony/Uhg.

Quantification of GABAergic neurons in intermediate layers of SC:

"GAD67-GFP"; B6.CG-Gad1TM1Tama.

In vivo electrophysiology:

"Rbp4-Cre-ChR2-EYFP"; crossbreed between "Rbp4-Cre" x "Ai32" (B6.FVB/CD1-Tg (Rbp4-Cre)KL100Gsat/Mmucd x B6.129S-Gt(ROSA)26Sortm32(CAG-COP4*H134R/EYFP).

### Virus-mediated labeling and intersectional strategies

*Retrograde labeling*: To identify input regions to the LSC (Fig 3A), we targeted the retrograde rAAV-TdTomato [30] into the whisker-sensitive region of the LSC (Fig 2). To delineate SC projection neurons (Td-Tomato) from L6 neurons, these injections were done in Ntsr1-EYFP mice, in which L6 but not L5 neurons express EYFP [54,55].

*Cortico-collicular bouton labeling*: To label MC and BC boutons in SC, we injected AAV1/2-CAG-SyPhy-EGFP and AAV1/2-CBA-SyPhy-mOrange [44] into the cortical projection sites (BC and MC, respectively) in the same animal (S2A Fig). The diameters of synaptic boutons labeled with Synaptophysin-EGFP and Synaptophysin-mOrange in SC were measured as maximal projection area [32,56] from confocal microscopy images using an A1+ microscope and NIS elements (Heidelberg Nikon Imaging Center).

**Table 1. Resources table.**

| Reagent or Resource | Source | Identifier |
|---|---|---|
| Antibodies | | |
| primary antibody rabbit IgG anti-NeuN | Merck Chemicals GmbH | ABN78 |
| primary antibody rabbit IgG anti-GABA | Sigma-Aldrich | A2052 |
| secondary antibody goat anti-rabbit IgG (H+L) Alexa 647 | Thermo Fisher Scientific/ Invitrogen | A32733 |
| Virus strains | | |
| PENN AAV hSyn Cre WPRE hGH | Addgene | #105553-AAV1 |
| AAV2-hSyn-DIO-EGFP | Addgene | #50457-AAV2 |
| AAV2hSyn-DIO-mCherry | Addgene | #50459-AAV2 |
| AAV8-hSyn Con/Fon EYFP | Addgene | #55650-AAV8 |
| AAV2-Ef1a-fDIO mCherry | Addgene | #114471-AAV2 |
| AAV1-EF1a-Flpo | Addgene | #55637-AAV1 |
| AAVrg-CAG-hChR2-tdTomato | Addgene | #28017- retrograde |
| AAV1/2-CAG-SyPhy-EGFP | Thomas Kuner Lab | N/A |
| AAV1/2-CBA-SyPhy-mOrange | Thomas Kuner Lab | N/A |
| Experimental models: organisms/strains | | |
| (B6.FVB(Cg)-Tg/(Ntsr1-Cre)GN220Sat/Mmucd x B6.129S-Gt(ROSA)26Sortm32(CAG-COP4*H134R/EYFP). | GENSAT Project at Rockefeller University Jackson Laboratory Created by: Hongkui Zeng | MGI:4358487 (MMRRC: 030648) MGI:5013789 |
| "Wild type"; C57BL/6N | Janvier | C57BL/6NRj Mouse - Janvier Labs (janvier-labs.com) |
| B6-Gad1tm2(Cre)Mony/Uhg | Hannah Monyer | MGI:4830465 |
| B6.CG-Gad1TM1Tama | Created by: Nobuaki Tamamaki | [53] |
| (B6.FVB/CD1-Tg(Rbp4-Cre)KL100Gsat/Mmucd x B6.129S-Gt(ROSA)26Sortm32(CAG-COP4*H134R/EYFP) | GENSAT Project at Rockefeller University Jackson Laboratory Created by: Hongkui Zeng | MGI:4367067 (MMRRC:031125) MGI:5313002 |
| Software and algorithms | | |
| Matlab 2018b | Mathworks | https://de.mathworks.com/products/matlab.html |
| Spike 2 | CED | https://ced.co.uk/es/products/spkovin |
| ImageJ/Fiji | NIH | https://imagej.net/software/fiji/downloads |
| Canvas X | ACD systems | |
| Microsoft office | Microsoft | www.microsoft.com |
| Kilosort 2.0 | Github | https://github.com/MouseLand/Kilosort |
| Phy | The Cortical Processing Laboratory at UCL | https://github.com/cortex-lab/phy.git |
| Bonsai | Bonsai foundation CIC | https://bonsai-rx.org/ |

*Trans-synaptic anterograde and intersectional labeling*: To label input-defined recipient neurons, we made use of the ability of AAV1-Cre and AAV1-Flpo particles to jump to post-synaptic neurons via vesicle release [4,57,58]. We adopted this strategy to reveal different subsets of SC neurons defined by their monosynaptic input from whisker-related projection sites of interest (MC, BC, Bs) and to characterize intersections between overlapping cell populations, by using different combinations of anterograde and reporter viruses (Table 2). For example, the RNs contain a subpopulation of inhibitory neurons (iRNs), and we employ intersectional approaches based on the differential labeling of neurons, which fulfill different conditions. For RNs to be labeled, they need to receive input from an area of interest (for example, BC). For the iRNs to be labeled in a different color, they need to fulfill both, the previous condition, AND the additional condition to express cre (from the GAD-Cre line). We used the following strategies to label: (1) *Recipient neurons (RNs)*; (2) *Cortical and brainstem RNs*; (3) *Inibitory recipient (iRNs)*; and (4) *Convergence neurons (CVG-RNs)* as follows:

**Table 2. Viruses used for the different targeting strategies in this study.**

| Virus | Virus short name | Titer in viral genomes per milliliter [vg/ml] | Supplier | Expression |
|---|---|---|---|---|
| PENN AAV hSyn Cre WPRE hGH | AAV-1-Cre | $1.8 \times 10^{13}$ | Addgene #105553-AAV1 | Cre |
| AAV2-hSyn-DIO-EGFP | DIO-EGFP | $3 \times 10^{12}$ | Addgene #50457-AAV2 | Cre-dependent EGFP |
| AAV2hSyn-DIO-mCherry | DIO mCherry | $4 \times 10^{12}$ | Addgene #50459-AAV2 | Cre-dependent mCherry |
| AAV8-hSyn Con/Fon EYFP | Con/Fon EYFP | $1 \times 10^{13}$ | Addgene #55650-AAAV8 | Cre- and Flpo-dependent EYFP |
| AAV2-Ef1a-fDIO mCherry | fDIO mCherry | $1.1 \times 10^{13}$ | Addgene #114471-AAV2 | Flpo-dependent mCherry |
| AAV1-EF1a-Flpo | AAV1-Flpo | $7 \times 10^{12}$ | Addgene #55637-AAV1 | Flpo |
| AAV1-Ef1a-fDIO EYFP | fDIO EYFP | $2.1 \times 10^{13}$ | Addgene #55641-AAV1 | Flpo-dependent EYFP |
| AAVrg-CAG-hChR2-tdTomato (AAV Retrograde) | rAAV-TdTomato | $7 \times 10^{12}$ | Addgene #28017-AAVrg | tdTomato |
| AAV1/2-CAG-SyPhy-EGFP | AAV-SyPhy | Not determined | T. Kuner Lab | EGFP |
| AAV1/2-CBA-SyPhy-mOrange | AAV-SyPhy | Not determined | T. Kuner Lab | mOrange |

1. *Recipient neurons (RNs)*: Cocktails of AAV1-Cre (*trans*-synaptic) + AAV2-DIO-mCherry (1:1) were injected into the projection sites of interest (MC, BC, Bs) in separate mice. A second injection of the Cre reporter virus AAV2-DIO-EGFP was targeted to SC to reveal RNs (Fig 4A).

2. *Cortical and brainstem RNs*: Same as RN but injecting MC and BC in the same mouse with a cocktail of AAV1-Flpo + AAV2-fDIO-mCherry (1:1), and Bs with a cocktail of AAV1-Cre + AAV2-DIO-EGFP (1:1). SC was injected with a cocktail of AAV2-fDIO-mCherry + AAV2-DIO-EGFP (1:1) to reveal cortical and brainstem RNs in different colors (Fig 4F).

3. *Inhibitory recipient neurons (iRNs)*: Same as RN approach but in GAD-Cre mice [34] (Fig 5B), using a cocktail of AAV1-Flpo + AAV2-fDIO-mCherry (1:1) for the projection sites of interest (MC, BC, Bs). SC was targeted with a cocktail of AAV8-Con/Fon-EYFP and AAV2-fDIO-mCherry (1:1). AAV8-Con/Fon-EYFP serves as a reporter virus to express EGFP when Cre and Flpo recombinases are coexpressed and thus labels iRNs, while AAV2-fDIO-mCherry labels RNs.

4. *Convergence neurons (CVG-RNs)*: AAV1-Flpo + AAV2-fDIO-mCherry (projection site 1) and AAV1-Cre + AAV2-DIO-mCherry (projection site 2) cocktails (1:1) were injected into wild-type mice, and an AAV8-Con/Fon-EYFP + AAV2-fDIO-mCherry cocktail (1:1) into SC to visualize convergent RNs (EYFP, co-innervated from projection sites 1 + 2; Fig 6A) and RNs (mCherry, innervated from projection site 1 only). See S5 Fig for controls of the conditional expression systems. Virus expression time was between 28 and 30 days.

## Stereotaxic surgeries

Mice were placed in a stereotaxic frame (Kopf Instruments) and anesthetized with 1.2 to 2 vol % isoflurane in medical oxygen at a flow rate of 0.8 l/min while keeping body temperature at 39°C. Carprofen (CP-Pharma) was administered subcutaneously (5 mg/kg) and lidocaine (Xylocaine 1%, Aspen Pharma) was injected under the scalp and around ear bars for local

**Table 3. Coordinates and injection volumes for stereotaxic AAV injections.**

| Injection site | Injection volume [nl] | AP [mm] | ML [mm] | DV [mm] |
|---|---|---|---|---|
| MC | 200 | 1 | 1 | −0.6 to −0.8 |
| BC | 200 | −1 | 2.95 | −0.6 to −0.8 |
| Bs (Sp5) | 200 | −5.9 | 1.7 | −4 to −4.2 |
| SC | 300 | −3.6 | 1.3 | −2 / −1.8 / −1.6 (3 injections, 100 nl each) |
| SC (retrograde experiments) | 200 | −3.6 | 1.3 | 2 |

anesthesia. Eyes were covered with Bepanthen ointment (Bayer) to keep them lubricated during the surgery. After ensuring the absence of tail- and toe-pinch reflexes, the skin was opened with a midline incision, the periosteum and aponeurotic galea were removed to visualize the reference points bregma and lambda, and the head was aligned to the stereotaxic frame. Small craniotomies were drilled above the injection sites and viral particle solutions were injected with calibrated glass micropipettes (Blaubrand; IntraMARK) at the following coordinates relative to bregma (AP, ML) and pia (DV), summarized in Table 3.

After viral particle injections, the pipette was left in place for 10 min to allow diffusion and was then slowly retracted. Finally, the skin above the skull was sutured with silk (PermaHand, 6–0; Ethicon), and the animal was transferred to its home cage.

### Slicing and immunolabeling

After virus expression, mice were exposed to a lethal dose of Ketamine (80 mg/kg) and Xylazine (10 mg/kg) and transcardially perfused with 4% paraformaldehyde (PFA) in PBS. The brain was removed and postfixed for 12 h in 4% PFA at 4°C. The same procedure was applied for GABA immunofluorescence labeling except using 2% glutaraldehyde and 2% PFA in PBS. Brains were sliced (50 μm) with a vibratome (Thermo Scientific Microm HM650V).

### NeuN immunofluorescence staining

SC slices were incubated in blocking solution (PB 0.1 M, with 4% normal goat antiserum (NGS, Biozol) and 0.4% Triton X-100 (Roth)) for 2 h, then washed 5 times in PB 0.1 M. Subsequently, slices were treated with primary antibody rabbit IgG anti-NeuN (Merck, ABN78) diluted in PB 0.1 M 1:500 with 1% NGS for 48 h at 4°C. Finally, after washing again 5 times with PB 0.1 M, slices were incubated for 2 h at room temperature in secondary antibody goat anti-rabbit IgG (H+L) Alexa 647 (Invitrogen, A32733) diluted 1:500 in PB 0.1 M containing 0.3% Triton X-100 and 3% NGS.

### GABA immunofluorescence staining

SC slices were preincubated in PB 0.1 M containing 1% sodium borohydride for 15 min to remove glutaraldehyde autofluorescence. Subsequently, SC slices were washed 3 times in PB 0.1 M and then incubated in a blocking solution (4% NGS and 0.3% Triton X-100 in PB 0.1 M) for 30 min. We then washed the slices 5 times in PB 0.1 M and incubated them with primary antibody rabbit IgG antiGABA (Sigma A2052) diluted in PB 0.1 M 1:1,000 with 1% NGS overnight at 4°C. The following day, we washed 5 times with PB 0.1 M and incubated them for 1 h at room temperature in secondary antibody goat anti-rabbit IgG (H+L) Alexa 647 (Invitrogen, A32733) diluted 1:1,000 in PB 0.1 M containing 1% NGS. Finally, slices were washed 5 times in PB 0.1 M and mounted with a mounting solution (Mowiol 4–88, Sigma-Aldrich).

## Microscopy

Injection sites, labeled cells in SC, and axonal outputs were imaged with a fluorescent microscope (Leica DM6000) outfitted with an automated stage to generate mosaics (Andor Neo camera, SCC02104) and laser lines 365, 470, 505, 530, and 625 nm. The objective HC PL APO 10×/0.40Dry was used to image the injection site and labeled cells in SC (voxel size $2 \times 2$ μm). We used the objective HC PL APO 20×/0.70Dry (voxel size $1 \times 1$ μm) for imaging RN axonal outputs. Varicosities, retrogradely labeled cells, and immunolabeled cells were imaged with a confocal microscope (Nikon A1R or C2, Nikon Imaging Center Heidelberg, Germany) outfitted with an automated mosaic stage and laser lines 405, 488, 561, and 640 nm. Objectives used: Nikon Plan Apo 20× NA 0.75 Nikon Plan Fluor 40× NA 1.30 oil immersion, Nikon Apo λS 60× NA 1.40. Slices were scanned in sequential mode; scanner zoom: 1× to 3×; resolution of individual tiles: $2,048 \times 2,048$ pixels. The laser power, photomultiplier gain, and pinhole (20 to 60 μm) were adjusted so that the signal was not saturated.

## Cell counting

Fluorescence image series of SC were loaded into Fiji and manually rotated to align all slices along the midline. From each image series, we selected the section that matched the most rostral part of SC (AP: −3.08 mm) in Paxinos' atlas [59], so that all series contained the same domain (AP: −3.08 to −4.28 mm). To select the most rostral section, we compared the shape and size of the hippocampus, optic tract, anterior pretectal nucleus, medial geniculate nucleus, and SC and its brachium anatomical references against Paxinos' −3.08 mm slice. The resulting SC series were processed in Fiji as follows. First, as a measure of background, we took the average pixel intensity of 3 different regions in the non-recipient area and subtracted this average value from each section in the series. Images were then contrast enhanced with an exponential transformation (Fiji "Exp" function), and brightness was manually restored by scaling intensity values by a factor between 1.2 and 2. Labeled neurons were detected using Fiji 3D Object Counter with 2 user parameters: gray-value threshold and a minimum-size filter, which we set to 30 pixels to exclude non-somatic signals. The distributions of detected neurons across the rostro-caudal (AP) SC axis are shown in 13 bins of 100 μm from −3.08 to −4.28 mm.

To estimate the proportion of GABAergic neurons in the intermediate layer of LSC, confocal image series of double-immunolabeled slices (GAD67 and NeuN) were analyzed with Fiji's "Cell Counter" plugin. GAD67- and NeuN-labeled somata were counted in 3 segments of $200 \times 200$ μm per image within the intermediate layer ($n = 10$ slices). We excluded marginal slices when the cell number did not satisfy a minimum $N$ sample criterion to detect the proportion difference with a statistical power of 0.8 β.

## RN distribution analysis

All analyses were done using MATLAB (R2018b). For comparisons across different RNs, neuron counts were normalized to the maximum value across sections within the same brain. Normality of the distributions was tested with one-sample Kolmogorov–Smirnov test and differences between the distributions were tested with Wilcoxon rank sum test. Mean values were reported with SEM (error bars or plus/minus) and median values with Q1 and Q3, or interquartile range (IQR). To compare distributions among different RN groups, a non-parametric analysis of variance was performed (Kruskal–Wallis and multiple comparisons test, using the Bonferroni method).

**Table 4. Axon count matrices.**

| RN output type | Diencephalic axon map | Brainstem axon map |
| --- | --- | --- |
| MC-RN | 13 slices × 15 nuclei | 11 slices × 7 nuclei |
| BC-RN | 13 slices × 15 nuclei | 14 slices × 7 nuclei |
| Bs-RN | 12 slices × 15 nuclei | 13 slices × 7 nuclei |

## Axon maps

Fluorescent axonal projections from RNs were reconstructed in Canvas X (version 10, ACD system). Coronal images were desaturated, inverted, and manually registered to the corresponding atlas section [59]; see schematics in Figs 8A, 8B and S7. We considered axonal signals in the AP regions between −1.22 and −2.54 mm (diencephalon) and −5.4 and −6.4 mm (brainstem). The resulting axon counts were organized into 2 matrices (for diencephalon and brainstem, respectively; Table 4) per input-defined RNs. Each matrix was $m \times n$, where $m$ is the number of 100-μm bins in the AP dimension and $n$ is the number of nuclei analyzed; 15 for diencephalon and 7 for brainstem.

Relative axon counts (RACs, normalized counts) were calculated for each pathway across the AP axis based on their maximum count as follows:

$$RAC = \frac{counts}{max(counts)}.$$

RAC maps were visualized using MATLAB's built-in function contour ("Fill", "on") and are shown in descending order, taking MC-RNs as reference. Finally, to compare the projection patterns among different RN groups, we summed the axonal counts per nuclei across the antero-posterior ($m$) dimension and calculated the z-score of the total axonal count per nucleus ($n$) (Fig 8B).

## In vivo electrophysiological recordings

**Awake recordings.** Mice between 8 and 12 weeks ($n = 12$) were recorded on a cylindrical treadmill, based on the model established in [60], consisting of a 15-cm diameter foam roller mounted on a custom built low friction rotary metal axis, attached to 2 vertical posts. Prior to whisker stimulation experiments, a head plate was implanted through stereotaxic surgery following the preparation procedures described in the section for viral injections. After exposing the skull, a small craniotomy was made above the recording site in LSC (AP: −3.6; ML: 1.25 to 1.3; DV: −1.65 to −2.1). A plastic ring was cemented (Paladur, Kulzer, GmbH) around the craniotomy to create a small ringer reservoir for the reference electrode. The well was covered with silicone elastomer (Kwik-Cast, World Precision Instruments) until the experiment. Next, a polycarbonate two-winged head plate was cemented onto the skull with dental cement (Super-bond, Sun Medical). Habituation began 5 days after the surgery and lasted for 3 days. Animals were head-fixed on the treadmill apparatus with the head plate. Habituation sessions lasted for approximately 60 min during which mice freely walked on the cylindrical treadmill and were fed sweetened condensed milk as reward. Recording sessions were done on the following day and lasted between 20 and 30 min. The protective silicone was removed and silicon probes were lowered to the SC. Activity was recorded using an electrophysiology system composed of a multielectrode silicon probe (sharpened ASSY-77 E1, Cambridge Neurotech); a 64-channel amplifier chip (RHD2164) to a USB-2 interface board (RHD-EVAL, Intan Technologies, California, US); and signals were fed into Bonsai [61] Intan RHD library. Neural activity was recorded at 30 kHz using a 0.1 Hz to 15 kHz bandpass filter. The resulting binary

file was fed to Kilosort, a MATLAB-based semi-automated spike-sorting software [62]. The. bin files and the E1-probe channel map in µm were input to Kilosort with the default parameters and the results were curated with Phy. Single units with <3% refractory period (1.5 ms) violations and a baseline spike rate >0.1 Hz were included for subsequent analysis.

**Whisker stimulation.** Whiskers were stimulated with airpuffs, delivered via an air tube placed behind the whiskers of the animal to prevent stimulation of any other body part (face, ears, or eyes). The tube was connected to an air regulator module from Modular Electronics for Cell Physiology (Max-Planck-Institute for Medical Research, manufactured by Sigmann Elektronik, GmbH, Germany) and set to 2 Bar. Airpuffs were controlled with Bonsai [61] and consisted of 100 ms airpuffs at random intervals between 1 s and 100 s.

**Anesthetized recordings.** Preparations and recordings were done in 8- to 12-week-old Rbp4-Cre-ChR2-EYFP mice ($n = 4$), which is crossbred between "Rbp4-Cre" × "Ai32" to specifically express ChR2 in L5 PT neurons [63]. Mice were anesthetized with a mixture of 5% urethane solution (IP, 1.3 g/kg body weight) and 1% isoflurane in medical degree oxygen, applied via an inhalator mask. After incision, bregma and lambda were revealed and aligned using a micro-manipulator (Luigs-and-Neumann) to drill craniotomies above SC (AP: −3.6, ML: 1.3), BC (AP: −1, ML: +3), and MC (AP: +1, ML: +1). SC activity was recorded using an electrophysiology system composed of a multielectrode silicon probe in AP: −3.6, ML: +1.3 DV: −1.6 to −2 mm (sharpened ASSY-77 E1, Cambridge Neurotech); a 64-channel amplifier chip (RHD2164) to a USB-2 interface board (RHD-EVAL, Intan Technologies, California, US); and the Spike2 (version 9.14) recording suite with Intan Talker module (Cambridge Electronic Devices, Cambridge, UK). Neural activity was recorded at 30 kHz using a 100 Hz to 10 kHz bandpass filter. The resulting file (.smrx) was converted into a binary file (.bin) to feed it to Kilosort, a MATLAB-based spike-sorting software [62]. The file conversion consisted of reading the channels in the.smrx file, transforming them into unsigned 16-bit integers (uint16) values from the 16-bit depth analog-to-digital converter (ADC), and writing them into a.bin file. The.bin files and the E1-probe channel map in µm were input to Kilosort with the default parameters, and the results were curated with Phy. Single units with <3% refractory period (1.5 ms) violations and a baseline spike rate >0.1 Hz were included for subsequent analysis.

**Optogenetic stimulation.** An optical fiber was placed at an angle of approximately 45˚, at approximately 100 µm from the pia to optogenetically activate L5 neurons in MC or BC. Optogenetic stimulation pulses were controlled by a stimulation protocol prepared in Spike2 through interface hardware (Power1401, Cambridge Electronic Design, Cambridge, UK). Single 5 ms laser pulses (20 to 50 mW/mm$^2$ at the fiber tip) were delivered every 5 s using a custom-build laser setup containing a 488-nm solid-state laser (Sapphire, Coherent, Germany, maximum power 22 mW); an ultrafast shutter (Uniblitz, Rochester, NY, USA); and a collimator to focus the laser beam into the optical fiber (inner diameter 400 µm, NA 0.48, Thorlabs, USA). After stimulating L5 in one cortex, the optical fiber was moved to the other, for example, MC and then BC.

**Cell responsiveness.** *Whisker responses*: We considered a single unit to be whisker responsive if the median spike count across trials within a 30-ms window (−20 to −50 ms) prestimulus was significantly different from the median spike count within an identical-length window (20 to 50 ms, α = 5%, two-sample Wilcoxon test). The modulation index (MI) was computed using Cell Explorer [27] as follows: $MI = (R + C)/(R - C)$, where $R$ and $C$ are firing rates in the response and spontaneous windows, respectively [64]. A given unit could have MI values ranging from −1, when $R = 0$ & $C \neq 0$; to 0, when $R = C$; and to 1, when $R \neq 0$ & $C = 0$.

*Optogenetic responses*: We considered 2 equal-length time windows in which we counted spikes before the onset of the optogenetic stimulus (spontaneous, −18 to −6 ms) and after the

optogenetic stimulus onset (evoked, 6 to 18 ms) for each trial and for each unit. We computed the z-scores of evoked counts using the mean and standard deviation of spontaneous counts and compared evoked z-scores against a 1.96 z-threshold ($\alpha$ = 5%). Trials that crossed the z-threshold were considered significantly responsive and were assigned a logical true value. We then computed the proportion of responsive (successful) trials per unit. In order to state significance for the response proportion, we estimated the necessary trial number (N) to achieve a statistical power of 0.8 with proportion values $p$ and hypothesis $p_0 = 1\text{-}p$ using MATLAB's test of the "N" parameter (trials) for a binomial distribution (sampsizepwr). For $p \approx 1/3$, $N \approx 40$, which is our experimental trial number. Additionally, we compared the unit's trial crossing proportions with increasing thresholds ($\alpha$ = 5%, one-tailed $t$ test, ttest in MATLAB) and visually searched for a region where the cardinality of included units stabilized. With an increasing threshold, the number of included units decreases until the responsive units would be found. We chose $p_0 = 1/3$ as an optimal inflection point, minimizing the threshold and maximizing the responsive unit cardinality.

## Supporting information

**S1 Fig. LSC-projecting neurons in the MC (M1 and M2). Related to Fig 3.** (**A**) Retrograde mCherry-labeled LSC-projecting neurons (red) in the MC extend across M1 and M2. (B) Higher magnification confocal image showing MC LSC-projecting neurons (red) in the motor cortex from pia to wm. LSC, lateral SC; MC, motor cortex; wm, white matter.
(TIF)

**S2 Fig. MC and BC boutons in SC. Related to Fig 3.** (**A**) Schematic of dual injections of AAVs encoding for synapse-specific fluorescent fusion proteins (Synaptophysin-mOrange, Synaptophysin-EGFP; [1,2]) to label MC (mOrange) and BC (EYFP) boutons in SC. (**B**) Example confocal images of coronal SC slices with fluorescently labeled BC (left, synaptophysin-EGFP) and MC (right, synaptophysin-mOrange) boutons. (**C**) Example of higher magnification confocal image showing BC boutons in the intermediate layers of SC. (**D**) Normalized distribution of MC and BC bouton diameters in SC. Boxplots of bouton diameters, median (line in box), IQR (first to third quartile, boxes) (BC = 1.15 μm, $n$ = 100; MC = 1.38 μm, $n$ = 44) and IQRs (BC = 0.50 μm; MC = 0.60 μm). * represents $p < 0.01$; B: Wilcoxon rank sum; exact $p$-values in S1 Table. The data for S2D Fig can be found at: https://doi.org/10.11588/data/DNOSZG. BC, barrel cortex; Int, intermediate layer; IQR, interquartile range; MC, motor cortex; SC, superior colliculus; Sup, superficial layer.
(TIF)

**S3 Fig. Trans-synaptic labeling of MC RNs in the intermediate gray and white layers [3], equivalent to SC intermediate layers. Related to Fig 4.** (**A**) SC coronal slice showing *trans*-synaptically EGFP-labeled MC-RNs in the intermediate layers of SC with overlaid anatomical borders from Paxinos mouse brain atlas [3]. (**B**) Same as (**A**) at higher magnification. csc, commissure of the superior colliculus; DpG, Deep gray layer; DpWh deep white layer; InG, Intermediate gray layer; InWh, intermediate white layer; MC, motor cortex; Op, optic nerve of the superior colliculus; PAG, periaqueductal gray; RN, recipient neuron; SC, superior colliculus; SuG, superficial gray layer; Zo, Zonal layer.
(TIF)

**S4 Fig. GABA immunostaining of Con/Fon labeled cells in the GAD-Cre mouse line. Related to Fig 5.** Example experiment in which MC-iRNs were labeled by injecting AAV1-Flpo in MC, and AAV8-Con/Fon-EYFP in LSC of GAD-Cre mice. Slices with *trans*-synaptically labeled iRNs (green, Confon-EYFP) were counterstained against GABA (red,

immunostaining with Alexa 647; see Materials and methods). All inspected iRNs were GABA positive. (**A**) Example confocal image showing iRNs (green) and GABA immunostain (red). Left: Overlay of red and green channel, showing GABA-positive iRNs in yellow (arrows). Middle: Green channel, showing iRNs (green). Right: Red channel, showing GABA-positive somata (red). (**B**) Same as in (**A**) for a different iRN at higher magnification. iRN, inhibitory RN; LSC, lateral SC; MC, motor cortex.
(TIF)

**S5 Fig. Controls for conditional expression of trans-synaptic reporter viruses in LSC. Related to Figs 4–6.** (**A**) Upper row: Conditional expression of the reporter DIO was tested by injecting AAV2-DIO-mCherry either alone (left) or in combination with AAV1-Cre (right) into the LSC. Bottom row. Examples of corresponding fluorescence images of the LSC. Conditional expression was only observed in combination with AAV1-Cre (right, red neurons). No leak expression of the reporter virus was detected. (**B**) Same as (**A**) but for Flpo-dependent reporter virus. No leak expression of the reporter virus was detected. (**C**) Upper row: Conditional expression of the reporter virus AAV8-Con/Fon-EYFP was tested by LSC injections of AAV8-Con/Fon-EYFP either alone or in different combinations with *trans*-synaptic viruses: AAV1-Cre and AAV1-Flpo. Bottom row: Examples of corresponding fluorescence images of the LSC. Conditional expression was only observed for the last combination AAV8-Con/Fon-EYFP and AAV1-Cre and AAV1-Flpo (green neurons); no leak expression of the reporter virus was detected.
(TIF)

**S6 Fig. Proportion of GABAergic neurons in intermediate layers of the LSC. Related to Fig 5.** Proportion of GABAergic GAD67-GFP positive neurons with respect to immunostained NeuN neurons in the intermediate layer of the LSC. Using a GAD-GFP mouse line, in which GABAergic neurons are labeled with GFP, neurons were counterstained with NeuN-Alexa 647 (Fig 5). The plot shows the proportion of GABA and NeuN neurons along the rostro-caudal LSC axis. The data for S6 Fig can be found at: https://doi.org/10.11588/data/DNOSZG
(TIF)

**S7 Fig. Axonal projections and varicosities of MC-RNs in diencephalic and brainstem target nuclei. Related to Fig 8.** (**A**) Desaturated and inverted fluorescence images at 2 different rostro-caudal coordinates in diencephalic regions (top and middle) and brainstem (bottom), showing MC-RN axons (black arrowheads) at low magnification. Roman numbers enumerate the nuclei shown in (**B**). (**B**) Confocal images show examples of axons in diencephalic nuclei CM/CL/PC MD, PO, VM, PF, and ZI and in brainstem nuclei, Gi and 7N. Roman numbers indicate the nuclei shown in (**A**). Small square box indicates the region selected for magnification. Insets show a high magnification of axons and varicosities (white arrowheads), scale bar = 5 μm. CL, centrolateral nucleus; CM, centro-medial nucleus; Gi, gigantocellular reticular nucleus; MC, motor cortex; MD, mediodorsal nucleus; PC, precentral; PF, parafascicular nucleus; PO, posterior nucleus; RN, recipient neuron; nucleus VM, ventromedial nucleus; ZI, zona incerta; 7N, Facial nucleus.
(TIF)

**S8 Fig. LSC-iRN projections to diencephalic regions. Related to Fig 8.** (**A**) Desaturated and inverted fluorescence images at 3 different rostro-caudal coordinates (from left to right) showing MC-, BC-, and Bs-iRNs outputs (top to bottom). Black arrowheads indicate iRN axons. (**B**) Left: Exemplary desaturated and inverted fluorescence images at 2 different rostro-caudal coordinates showing MC-iRNs outputs in diencephalic nuclei (top and bottom). Black arrowheads indicate axons. Roman numbers enumerate the nuclei shown in high magnification. ZI,

zona incerta. *Right*: High magnification confocal images show examples of MC-iRN axons in ETH PF, Po, and ZI. Small square box indicates the region selected for magnification Roman numbers indicate the nuclei shown left. Insets show a high magnification of axons and varicosities (white arrowheads), inset scale bar = 10 μm. APT, anterior pretectal area; BC, barrel cortex; Bs, brainstem; ETH, ethmoidal nucleus; iRN, inhibitory RN; LSC, lateral SC; MC, motor cortex; MD, mediodorsal nucleus; Pf, parafascicular nucleus; PO, posterior nucleus; RN, recipient neuron; VM, ventromedial nucleus; ZI, zona incerta.
(TIF)

**S9 Fig. RNs are confined to the lateral zone of SC. Related to Figs 4, 5 and 6.** (**A**) Experimental schematic of *trans*-synaptic labeling to validate sufficient spread of the reporter virus to label most of the RN population in the recipient zone in SC. A cocktail of AAV1-Cre + AAV1-Flpo and reporter AAV8-ConFon-EYFP was injected in the barrel cortex. In SC, 2 different reporters fDIO-mCherry and DIO-EGFP were injected into adjacent locations corresponding to the medial and lateral zone of SC, respectively. (**B**) SC coronal slice showing that BC-RNs were mostly labeled with EGFP in the lateral zone and very few BC-RNs were labeled by mCherry in the medial zone. BC, barrel cortex; RN, recipient neuron; SC, superior colliculus.
(TIFF)

**S10 Fig. Negatively modulated clusters in the whisker-sensitive region of the LSC. Related to Fig 2.** (**A**) Experimental schematic of whisker airpuff stimulation and silicon probe recording in LSC in awake mice. (**B**) Summary from 12 recordings (1,005 units, 8 mice) mapped onto SC outlines through trilateration in CellExplorer [4,5]. Each dot depicts the location of a unit; colors indicate the negative modulation strength upon whisker stimulation (see Materials and methods). See main Fig 2C for modulated units. The data for S10B Fig can be found at: https://doi.org/10.11588/data/DNOSZG. LSC, lateral SC; SC, superior colliculus.
(TIFF)

**S11 Fig. RNs, iRNs, and ratios, split by analyzed brains. Related to Fig 5.** (**A**) RN counts per brain per pathway (MC: red; BC: blue; Bs: red), mean (dashed line), and standard deviation (grey shaded area). (**B**) Same as in (**A**) but for iRNs. (**C**) Same as (**A**, **B**) but for the resulting ratio of iRNs/RNs. The data for S11A-S11C Fig can be found at: https://doi.org/10.11588/data/DNOSZG. BC, barrel cortex; Bs, brainstem; iRN, inhibitory RN; MC, motor cortex; RN, recipient neuron.
(TIFF)

**S12 Fig. RNs, CVGs, and ratios, split by analyzed brains. Related to Fig 6.** (**A**) RN counts per brain per pathway (brainstem and cortex: upper panel, red; and MC and BC: bottom panel, blue), means (dashed lines), and standard deviations (grey shaded areas). (**B**) Same as in (**A**) but for CVG-RNs. (**C**) Same as (**A**, **B**) but for the resulting ratio of CVGs/RNs. The data for S12A-S12C Fig can be found at: https://doi.org/10.11588/data/DNOSZG. BC, barrel cortex; MC, motor cortex; RN, recipient neuron.
(TIFF)

**S1 Table. Statistical comparisons per figure.**
(DOCX)

**S2 Table. Sample sizes per figure.**
(DOCX)

**S3 Table. SC unit population spiking characteristics.** Reported values are medians, Q1, and Q3. Related to Figs 2 and 7.
(DOCX)

# Acknowledgments

We thank Alexei Egorov (Heidelberg University) for helping with the rAAV-Td-Tomato experiments and Martha Bickford and her team (University of Louisville) for helping with the GABA immunolabeling. We also thank the Nikon Imaging Center at the University of Heidelberg for their help with microscopy.

# Author Contributions

**Conceptualization:** Jesús Martín-Cortecero, Alexander Groh.

**Data curation:** Jesús Martín-Cortecero, Emilio Ulises Isaías-Camacho, Berin Boztepe.

**Formal analysis:** Jesús Martín-Cortecero, Emilio Ulises Isaías-Camacho, Berin Boztepe.

**Funding acquisition:** Alexander Groh.

**Investigation:** Jesús Martín-Cortecero, Emilio Ulises Isaías-Camacho, Berin Boztepe, Katharina Ziegler.

**Methodology:** Jesús Martín-Cortecero, Emilio Ulises Isaías-Camacho, Katharina Ziegler.

**Project administration:** Katharina Ziegler, Alexander Groh.

**Resources:** Jesús Martín-Cortecero, Rebecca Audrey Mease.

**Software:** Emilio Ulises Isaías-Camacho, Rebecca Audrey Mease.

**Supervision:** Jesús Martín-Cortecero, Katharina Ziegler, Rebecca Audrey Mease, Alexander Groh.

**Validation:** Jesús Martín-Cortecero, Emilio Ulises Isaías-Camacho, Berin Boztepe, Alexander Groh.

**Visualization:** Jesús Martín-Cortecero, Emilio Ulises Isaías-Camacho, Berin Boztepe, Rebecca Audrey Mease.

**Writing – original draft:** Jesús Martín-Cortecero, Emilio Ulises Isaías-Camacho, Alexander Groh.

**Writing – review & editing:** Jesús Martín-Cortecero, Emilio Ulises Isaías-Camacho, Berin Boztepe, Katharina Ziegler, Rebecca Audrey Mease, Alexander Groh.

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
