## [Editor Report · Decision Letter 0]

13 Jan 2023

Dear Dr Groh, 

Thank you for submitting your manuscript entitled "Monosynaptic trans-collicular pathways for sensory-motor integration in the whisker system" for consideration as a Research Article by PLOS Biology.

Your manuscript has now been evaluated by the PLOS Biology editorial staff, as well as by an academic editor with relevant expertise, and I am writing to let you know that we would like to send your submission out for external peer review.

Once your full submission is complete, your paper will undergo a series of checks in preparation for peer review. After your manuscript has passed the checks it will be sent out for review. To provide the metadata for your submission, please Login to Editorial Manager (https://www.editorialmanager.com/pbiology) within two working days, i.e. by Jan 15 2023 11:59PM.

Kind regards,

Kris

Kris Dickson, Ph.D., (she/her)

Neurosciences Senior Editor/Section Manager

PLOS Biology

kdickson@plos.org

---

## [Decision Letter · Decision Letter 1]

15 Feb 2023

Dear Dr Groh,

Thank you for your patience while your manuscript "Monosynaptic trans-collicular pathways for sensory-motor integration in the whisker system" went through peer-review at PLOS Biology. Your manuscript has now been evaluated by the PLOS Biology editors, an Academic Editor with relevant expertise, and by several independent reviewers.

In light of the reviews, which you will find at the end of this email, we are pleased to offer you the opportunity to address the comments from the reviewers in a revision that we anticipate should not take you very long and should largely involve rewriting (though please do also note the viral injection concern from Reviewer 2). We will then assess your revised manuscript and your response to the reviewers' comments with our Academic Editor aiming to avoid further rounds of peer-review, although might need to consult with the reviewers, depending on the nature of the revisions.

**IMPORTANT - SUBMITTING YOUR REVISION**

PLEASE CAREFULLY AND FULLY ADDRESS ALL OF THE POINTS BELOW, INCLUDING THE RESUBMISSION CHECKLIST!

**RESUBMISSION CHECKLIST** 

IMPORTANT: FAILURE TO FULLY ADDRESS THE BELOW POINTS WILL DELAY FURTHER HANDLING OF YOUR SUBMISSION

Please make sure to read the following important policies and guidelines while preparing your revision. This includes information on:

*Published Peer Review*

*PLOS Data Policy*

We require that authors provide the individual numerical values that underlie the summary data displayed in their figure panels as this information is essential for readers to assess the analyses and to reproduce it.

1) Please provide a summary data file, deposited either as:

a. A Supplemental file (e.g. excel)

b. Deposition to a static site, like Zenodo, FigShare, OSF...

2) The summary data file needs to include the numerical values for all of the graphical files in both the main and supplemental figures.

3) Please ensure that figure legends in your manuscript include information on where the underlying data can be found (e.g. “The underlying data supporting Fig X, panel Y can be found in file Z.”).

4) Please also ensure that your supplemental data file/s has a legend.

*Blot and Gel Data Policy*

Sincerely,

Kris

Kris Dickson, Ph.D., (she/her)

Neurosciences Senior Editor/Section Manager

PLOS Biology

kdickson@plos.org

REVIEWS:

Reviewer's Responses to Questions

Do you want your identity to be public for this peer review?

Reviewer #1: Yes: Ehud Ahissar

Reviewer #2: No

Reviewer #3: Yes: Wolfger von der Behrens

Reviewer #1: The SC is often thought to be an "old cortex", that is, it is thought that it used to contain many functions later recapitulated by the neocortex. As such, the SC is at the midst of many sensory-motor pathways, that is, pathways connecting sensory information to motor actions. Also, the SC is subject to rich cortical control, possibly mediating the necessary coordination between the "new" and "old" sensory-motor hubs.

The current study by Martin-Cortecero et al sheds novel light on this evolutionary dialogue. Focusing on the input to the SC, in fact to the whiskers-related part of the SC, the study revealed that cortical neurons directly control the output neurons of SC, via converging inputs from at least two cortical areas - motor cortex and barrel cortex. This control affects the brainstem-SC-brainstem sensory-motor arc, as well as the outputs from SC to midbrain stations.

The study presents micro-anatomical evidence for this convergence in the whisker tactile modality by revealing the input/output organization of the lateral zone of the superior colliculus (SC) that contains the whisker-related pathways. The authors show that cortical and trigeminal inputs target the GABAergic and non-GABAergic recipient neurons of this zone, which provide the output directed to diencephalic and brainstem nuclei - the former including "high-order" thalamic nuclei involved in the processing of self-motion (PO, ZI, PF and VM) and the latter involving nuclei involved in the execution of self-motion (PnC, Gi and 7N).

The methodological basis of this study includes modern methods to achieve the goals set in the introduction. The authors used multiple fluorescent tracing strategies with a combination of classic tract-tracing and viral tracing methods. They mapped the input-output connectivity of the mouse SC on the single-cell level with transsynaptic tracing, used intersectional viral approaches and optogenetic-assisted electrophysiology, applied computational neuroanatomical methods to analyze the representative cortico-tectal pathways that span the rostral-to-caudal and medial-to-lateral extent of the cerebral cortex and SC. Obtained labels were overlapped by the respective atlas templates.

This study is timely and exciting - it reveals information that is crucial for understanding the operation and coordination among parallel sensory-motor arcs - those running primarily via the SC and their evolutionary competitors running via the neocortex. The conceptual basis of this work, the outlines of the approach, methods and analyses are appropriate for the objectives of the study. The MS is clearly and scholarly written.

There are several issues that require the attention of the authors.

Major:

The authors do not refer to the parallel-arcs context mentioned above, relating the SC-mediated and the neocortex-mediated sensory-motor arcs to each other. As a result, the paper lacks an important angle that seems to be crucial for understanding the meaning of their results. I thus strongly suggest to add it, at the outset, and to add interpretations of the data in that direction.

Two terminology/phrasing issues refer to the above:

a) sensorimotor. This term (or equivalently "sensory-motor") typically refers to the sensory-motor arc. That is, to the processing done in brain circuits eventually transfroming activity from the sensory end to the motor end. Here the authors use "sensorimotor" when referring to convergence of signals from sensory and motor cortices. This is confusing and I suggest to stick to the traditional usage of this term. Of course, traditions are in the eye of the beholder, thus if the authors do not agree with the above they do not have to adopt this recommendation. In any case, it should be noted in the paper that layer 5 neurons of both S1 and M1 convey motor-related signals.

b) upstream/downstream (e.g., Abstract: "The results …by which upstream neurons in motor- and somatosensory cortices are linked to specific downstream whisker circuits in the brainstem …") - the "upstream" and "downstream" affiliations imply some agreement about the direction of information/processing flow among these stations. However, this doesn't seem to be the case. I suggest a different terminology, respecting the complex interplay between the sensory-motor arcs and the loopy organization of processing (both between the arcs and between the sensory and motor ends).

And one additional suggestion:

c) one possible distinguishing function between SC- and cortex-related arcs is the primary focus of their corresponding brain-world loops - self-motion for SC-mediated loops and external objects for cortical-mediated loops. If the current results can be interpreted in a way supporting or disproving such distinction it is strongly suggested to add such interpretations. In any case - the distinction is worth mentioning.

Other comments

1) It may be useful if the authors designate the zone- and layer-specific nomenclature within the SC in their figures to facilitate more detailed referencing and precise quantification of the revealed inputs. The authors touched these issues when defining the three principal questions about the organization of transcollicular pathways. In setting out the first question (P.2, 2nd para), the authors referred to the problem of the zonality of the distribution of input structures in the SC when citing the article by Benavidez et al. (2021) and calling them "mesoscopic modules" that include the structures identified in the current study. It looks like the authors do not reject their significance, which allows the use of such nomenclature in this study.

2) In the subsection "Slicing and immunolabeling" (P. 31), the authors wrote that the brain was "postfixed for 12 h in 4% PFA at 4°C". It looks as not enough to prevent freezing damage of the tissue and to preserve its structural integrity during slicing. In the studies that use similar methods, effective postfixation of the mouse brain is achieved by using hypertonic sucrose [see, for example, papers by Wall et al. (2010) and Castro-Alamancos and Favero (2016)], or by postfixation in 4% PFA for 24-48 h at 4°C (Benavidez et al., 2021). If the authors used sucrose or other cryoprotectants in this study, it should be mentioned.

3) In figure 3E, the spatial position of the atlas templates needs to be corrected with respect to the midline and the scale.

4) The study of the LSC projection neurons in Sp5o looks quite reasonable because of its involvement into whisker circuitry which was previously revealed (Erzurumlu et al. 2010; Ebert et al., 2021). However, as it was shown 40 years ago by Huerta et al. (1983), other trigeminal nuclei, such as Principal and Interpolaris, have a larger number of neurons projecting to the SC. If the authors have the data about neuronal populations of other brainstem nuclei that project to the SC, it would be interesting to compare them with the Sp5o.

5) The MS contains a lot of standard and non-standard abbreviations. An abbreviation should appear at the first occurrence of the word, which is not always respected by the authors. For example, the abbreviation for zona incerta (ZI) appeared on P. 17, while these unabbreviated words are seen on P. 3 (3rd line from the bottom).

6) In the third horizontal row, fourth vertical cell of the Table 1: replace "mCHerry" by "mCherry".

7) P. 7, Line 7 from the top: "Exact N in Supplementary Table S2." should be moved to the Line 3 of P.7 because it is related to the legend for Fig. 3D, as it is mentioned in the Table S2.

8) The term "anterior-posterior" is used in the MS to designate coordinates and axes in many places in the text, in figures and figure captions and in Table 2. It should be replaced by the term "rostro-caudal", referring to the coordinates of the rodent brain.

Reviewer #2: The manuscript by Martin-Cortecero et al. entitled 'Monosynaptic trans-collicular pathways for sensory-motor integration in the whisker system' is thorough and relevant. The authors describe monosynaptic transcollicular pathways connecting motor and somatosensory cortices, as well as the trigeminal complex, with thalamus and zona incerta via long range excitatory and inhibitory connections. The novelty of the manuscript strives in the description of monosynaptic, somatosensory, GABAergic pathways, as well as the convergence of motor and sensory terminals on superior collicular projection neurons. The experiments are thorough and convincing. The manuscript is well written. The work reveals a novel loop of cortico-collicular-thalamic projections where sensory-motor information can be integrated. In addition it reveals a novel loop of cortico-collicular-zona incerta projections that is very likely to play an important role in the modulation of sensory-motor interactions.

Major comments:

My main concern is with the quantifications of anatomical distribution of projections and percentage of iRN and CVG-RN. I would need to be convinced that these quantifications are not biased by the viral injection location or the efficiency of the different viruses. Comparing results across mice and/or using different viruses would make the case stronger. Given the number and strength of experiments reported here, it might be enough to justify the value and report it as indicative.

Minor comments aiming to help with the clarity of the manuscript:

Title: Short titles are nice but, since S1 projects also to inferior colliculus and inferior colliculus also perform sensory-motor integration, this title is ambiguous. 

Abstract: 'we mapped the anatomical and functional…'. Not sure what it is meant by mapping the functional input-output properties since no function was assessed. Maybe more accurate to say 'physiological'?

Abstract: '…GABAergic neurons, which in turn give rise to hitherto unknown…' This sentence gives them impression that all GABAergic RN lead to long-rang connections. I would rephrase.

Introduction: while the idea of focusing on specific question and describing them in turn is very good, I would briefly introduce the same questions in the first paragraph of the introduction.

Consistency in naming: e.g. in the first paragraph of the introduction the question is about 'target in the diencephalon and brainstem, while in the first question t refers to projections to 'thalamus and brainstem'. Both are correct but maybe for the general reader it is worth including more details? For example to say first 'diencephalon (including thalamus)', and then use 'diencephalon' both times?. 

Introduction, question 2. Describe visual and somatosensory SC for a wider PloS Biology audience

Page 3, last paragraph: Is 'intersectional' a standard term? I am not sure what it means? Across SC sections? Histology?

Page 3, last paragraph: 'optogenetically-assisted electrophysiology', maybe use the more common term opto-tagging?

Page 3, last paragraph: the sentence 'suggesting trans-collicular circuits for cortical disinhibitory control of higher-order thalamus' is loaded with very specific information that has not ben previously mentioned. Maybe a bit of a mouthful for a general conclusion at the end of the introduction?

Figure 2A: like all the others schemes in this manuscript with the exception of those in Figure 1, this scheme is beautiful and very useful but very small!!! Maybe have the detailed scheme once enlarged and from then on a less realistic but larger scheme for MC, BC and Bs. For example by Figure 5A, we know what SC looks like and where it is and it would suffice to show just SC.

The same is true for most of the neuron target reconstructions, e.g. Fig 5D, 6B, 8A. One could cut the hippocampus and overlap the sections slightly.

Page 4, results, 1st paragraph: it would be useful to have the information on how many mice were used together with that of the number of recordings.

Figure 2C: I don't see the separation between positive and negative modulation referred to at the beginning of the results.

Figure 2B shows a clear bimodal distribution between fast and slow responses. All of them look positively modulated by the whisker at some point? Please explain in more detail in the text.

Page 4, last sentence: I get the meaning but I think the sentence needs rephrasing.

Page 7: Bs-projecting. I think this is the first time the abbreviation Bs is used other than the figure. 

Page 7, last paragraph. I am not sure I understand the new conclusion here? The tracing has ben done in the LSC, which is already identified as somatosensory. Do the authors refer to 'whisker SC' as a specific form of somatosensory?

Figure 4C and E do not seem to match well. In the Bs example in Figure 4c many neurons seem to be in the ventral layer.

Figure 4F; please include in the legend how many mice were used for co-labelling.

Page 9: the term trigemino-collicular is used in the titles and legends, but the term brainstem and abbreviaton Bs are used in most of the text. Is there a reason not to chose one?

Figure 5A, 22% of cells are reported as being GABAergic, but in the text it says 23%. Pleas round consistently.

Discussion, paragr. 2: 'midbrain and brainstem'. I assume that, if the focus is on to targets, 'diencephalon and brainstem' are meant?

Discussion, paragr. 2: '…fundamentally different from that found in other layered circuits - for example, compared to the canonical circuit scheme of the cortex, in which layer 4 neurons…'. I am not 'fundamentally different' is the right word. If the example would have been layer 6, the pattern wouldn't look so different. The long-range inhibitory projections are unusual but maybe not fundamentally different (see, as an example, the work on long-range inhibitory projections form the ZI to cortex by Schoeder et al. from the Letzkus lab in BioRxiv)

Reviewer #3: Martín-Cortecero and colleagues present a new pathway for whisker sensory-motor integration where long-range motor-sensory innervate in the lateral superior colliculus and from there inhibitory and non-inhibitory pathways target thalamic and brainstem nuclei.

This microanatomical study is very well performed and presents a somatosensory-motor circuit that has not been described before. The authors performed very well controlled tracing experiments by combining different viral systems for anterograde and retrograde labelling of neurons and their targets. They complement these experiments with awake recordings and optogenetic stimulation to demonstrate the functionality of the described circuits. The amount of data presented and the level detail is impressive and have to admit that I find it tremendously difficult to spot a weakness in this study or things to improve. 

I consider this study to be relevant for a broader readership as the described trans-collicular circuits are likely performing very critical sensory-motor integration that runs in parallel to the well studies and described thalamo-cortical loop. While present study was performed in the whisker system, there is good reason to assume that it is not limited to this modality but may be more generally relevant for other (multimodal) sensory-motor integration functions as well. 

Given the relevance and quality of the study I recommend it for publication without any reservations. However, I would like to urge the authors as a minor revision to make the full dataset including the raw data available through an appropriate repository such as Zenodo or similar.

---

## [Editor Report · Decision Letter 2]

25 Mar 2023

Dear Dr Groh,

Thank you for your patience while we considered your revised manuscript entitled "Monosynaptic trans-collicular pathways for sensory-motor integration in the whisker system" for publication as a Research Article at PLOS Biology. This revised version of your manuscript has been evaluated by the PLOS Biology editors and the Academic Editor..

Based on our Academic Editor's assessment of your revision, we are likely to accept this manuscript for publication, provided you satisfactorily address the data and other policy-related requests stated below.

In addition, we would like you to consider a suggestion to improve the title:

"Monosynaptic trans-collicular pathways link mouse whisker circuits to integrate somatosensory and motor cortical signals"

We expect to receive your revised manuscript within two weeks. 

*Published Peer Review History*

*Press*

Sincerely,

Ines

--

Ines Alvarez-Garcia, PhD

Senior Editor

PLOS Biology

ETHICS STATEMENT:

Many thanks for providing the ethics statement. Please add the license number.

DATA POLICY:

I have checked the data deposited in heiDATA and I am missing the data underlying the graphs shown in the following figures:

Fig. 2B and Fig. 7B

Please provide it or let us know where the data is located.

---

## [Editor Report · Decision Letter 3]

14 Apr 2023

Dear Dr Groh,

Thank you for the submission of your revised Research Article entitled "Monosynaptic trans-collicular pathways link mouse whisker circuits to integrate somatosensory and motor cortical signals" for publication in PLOS Biology. On behalf of my colleagues and the Academic Editor, Mathew Diamond, I am delighted to say that we can in principle accept your manuscript for publication, provided you address any remaining formatting and reporting issues. These will be detailed in an email you should receive within 2-3 business days from our colleagues in the journal operations team; no action is required from you until then. Please note that we will not be able to formally accept your manuscript and schedule it for publication until you have completed any requested changes.

PRESS

Sincerely, 

Ines

--

Ines Alvarez-Garcia, PhD

Senior Editor

PLOS Biology
